# LINC complex-Lis1 interplay controls MT1-MMP matrix digest-on-demand response for confined tumor cell migration

Elvira Infante[1], Alessia Castagnino[1], Robin Ferrari[1], Pedro Monteiro[1], Sonia Agüera-González[1], Perrine Paul-Gilloteaux [1,2], Mélanie J. Domingues[1], Paolo Maiuri [3], Matthew Raab[1], Catherine M. Shanahan[4], Alexandre Baffet[1], Matthieu Piel [1], Edgar R. Gomes [5] & Philippe Chavrier [1]

Cancer cells' ability to migrate through constricting pores in the tissue matrix is limited by nuclear stiffness. MT1-MMP contributes to metastasis by widening matrix pores, facilitating confined migration. Here, we show that modulation of matrix pore size or of lamin A expression known to modulate nuclear stiffness directly impinges on levels of MT1-MMP-mediated pericellular collagenolysis by cancer cells. A component of this adaptive response is the centrosome-centered distribution of MT1-MMP intracellular storage compartments ahead of the nucleus. We further show that this response, including invadopodia formation in association with confining matrix fibrils, requires an intact connection between the nucleus and the centrosome via the linker of nucleoskeleton and cytoskeleton (LINC) complex protein nesprin-2 and dynein adaptor Lis1. Our results uncover a digest-on-demand strategy for nuclear translocation through constricted spaces whereby confined migration triggers polarization of MT1-MMP storage compartments and matrix proteolysis in front of the nucleus depending on nucleus-microtubule linkage.

---

[1] Institut Curie, PSL Research University, CNRS, UMR 144, 26 rue d'Ulm, F-75005 Paris, France. [2] Institut Curie, Cell and Tissue Imaging Facility (PICT-IBiSA), 26 rue d'Ulm, F-75005 Paris, France. [3] IFOM, the FIRC Institute of Molecular Oncology, Via Adamello 16, 20139 Milan, Italy. [4] BHF Centre of Research Excellence, Cardiovascular Division, King's College, 125 Coldharbour Lane, SE5 9NU London, UK. [5] Instituto de Medicina Molecular, Faculdade de Medicina, Universidade de Lisboa, Avenida Professor Egas Moniz, 1649-028 Lisboa, Portugal. These authors contributed equally: Elvira Infante, Alessia Castagnino, Robin Ferrari, Pedro Monteiro. Correspondence and requests for materials should be addressed to E.I. (email: elvira.infante@kcl.ac.uk) or to P.C. (email: philippe.chavrier@curie.fr)

Recent studies revealed that limited deformability of the nucleus prevents constricted cell movement and that nuclear stiffness is a critical element for the ability of normal and cancer cells to migrate through confined extracellular matrix (ECM) environments[1–4]. Nuclear rigidity depends on lamin A (LMNA) levels, component of the nuclear lamina acting as a rigid and protective shell underneath the inner nuclear membrane[5,6]. Down-modulation of LMNA in cancer cells correlates with increased nuclear deformability and enhanced cell migration in confined environments by facilitating nucleus squeezing through ECM pores[1–4,7–9].

Also critical for metastasis is the capacity of cancer cells to remodel ECM barriers[10]. Invasion by carcinoma cells is potentiated by pericellular matrix proteolysis, executed by trans-membrane membrane-type 1 matrix metalloproteinase (MT1-MMP)[11,12]. MT1-MMP is up-regulated during tumor progression and its up-regulation predicts the invasive potential of cancerous breast lesions[13,14]. In 3D type I collagen network, pericellular ECM proteolysis is associated with the invasive cell protrusion ahead of the nucleus, and is reduced at the cell leading edge, involved in cell-matrix adhesion to support 3D migration[15,16]. With decreasing matrix pore size, cancer cell invasion critically depends on MT1-MMP surface expression to enlarge matrix pores[2,11]. Inhibition of MT1-MMP function impairs confined cell movement and correlates with increased nuclear deformation, nuclear envelope (NE) rupture and DNA damage[2,15,17].

Cancer cells adjust their levels of surface-exposed MT1-MMP through trafficking from late endosomal/lysosomal storage compartments[18]. Whether and how matrix porosity and cell confinement influence MT1-MMP surface localization remain unexplored. To address these outstanding questions, we used live cell imaging of breast carcinoma and fibrosarcoma cells invading through 3D collagen gels of controlled porosity. We report that invasion through small pore size collagen meshwork triggers an adaptive response with polarized centrosome-centered distribution of MT1-MMP-positive storage endosomes ahead of the nucleus and enhanced MT1-MMP-based pericellular proteolysis of confining collagen fibrils. In contrast, endosome polarization is lost and collagenolysis decreases in cells invading through a permissive large pore size collagen environment. Importantly, modulating LMNA levels with known consequences on nuclear stiffness impinges on MT1-MMP-positive endosome polarity and collagenolysis. We provide evidence that endosome polarization and MT1-MMP-dependent collagenolysis require integrity of the linker of nucleoskeleton and cytoskeleton (LINC) complex that connects the nuclear lamina to cytoskeletal elements in the cytoplasm and the dynein regulator Lis1 involved in nucleus-microtubule cytoskeleton linkage[19]. Our data support a model whereby focal MT1-MMP-mediated ECM proteolysis response is engaged by mechanical signals during confined migration to facilitate nuclear movement and promote tumor cell invasion.

## Results

**Confinement and nuclear stiffness regulate collagenolysis.** The morphology and collagenolysis activity of invasive MDA-MB-231 cells embedded in the 3D fibrillar type I collagen network were analyzed by staining for microtubules and a cleaved collagen neoepitope. After a short 2.5 hrs incubation, different cell morphologies were observed (Fig. 1a): (i) pre-invasive rounded cells with collagen degradation surrounding the cell edge, (ii) cells that initiated invasion as exemplified by limited collagen degradation track behind the cell and at the basis of the nascent protrusion ahead of the nucleus, (iii) fully invasive cells showing typical elongated mesenchymal organization with collagen

degradation in front of the nucleus and cleared collagen from the cell path probably through the action of collagenases, consistent with previous observations[20]. Importantly, when fluorescence intensity of cleaved collagen was measured along the long cell axis of several invasive cells and averaged, a robust pericellular collagenolysis in association with the bulky part of the cell anterior to the nucleus was observed, while collagen degradation was minimal at the cell front (Fig. 1b). In agreement with previous findings[2,3,17], we observed that $28 \pm 2.9\%$ of invasive cells in the 3D collagen gel presented various degrees of nuclear deformation as the nucleus moved through constricting spaces (Fig. 1d and Supplementary Fig. 1a). Moreover, inhibition of MMP activity upon GM6001 (GM) treatment, which reduced invasion speed in 3D collagen by ~2-fold and interfered with pericellular collagenolysis (Fig. 1e, f and Supplementary Fig. 1b), correlated with a strong enhancement of nuclear deformation (~55–60% deformed nuclei, see Fig. 1d and Supplementary Fig. 1c). Similarly, MT1-MMP silencing using small interfering (si)RNA led to a robust inhibition of pericellular collagenolysis and increased nuclear deformation (Fig. 1d, g and Supplementary Fig. 1b). Altogether, these findings indicated that surface-exposed MT1-MMP enabled efficient cell movement in confinement by mediating proteolysis anterior of the nucleus.

Physical features of the matrix and cellular parameters can affect cancer cell motility such as matrix porosity and nuclear stiffness, respectively[1–4]. We explored potential relations between these parameters and MT1-MMP-dependent pericellular collagenolysis. Matrix pore size was increased by reducing collagen polymerization temperature to 20 °C instead of 37 °C, the condition used so far, while keeping collagen concentration constant (2.0 mg/ml); this led to a ~2-fold increase in the distance between fibrils (Supplementary Fig. 1d, e)[2,21]. GM treatment did not result in a significant increase of nuclear deformation in cells invading through the gel of higher porosity (Fig. 1d). In addition, invasion speed in the large pore size gel was not affected by GM treatment contrasting with inhibition observed in the smaller pore size gel (Fig. 1e). These data indicated that in a permissive large pore size collagen environment causing reduced nuclear constriction, MT1-MMP was dispensable for invasion, in agreement with previous observation[2]. Strikingly, we found that invasion through the higher porosity gel correlated with a ~60% reduction of pericellular collagenolysis as compared to the small pore size collagen network and a lack of proteolysis anterior of the nucleus (Fig. 1b, c, f and Supplementary Fig. 1b). Similarly, pore size enlargement correlated with a significant decrease of collagenolysis by HT-1080 fibrosarcoma cells, while invasion speed was similar in large and small pore size gels (Supplementary Fig. 1f–h). These data indicated that modulation of cell confinement correlated with changes in MT1-MMP-dependent pericellular proteolysis of constricting collagen fibrils.

Next, the influence of lamin A expression levels, with well described relationship with nuclear stiffness and deformability was tested on MT1-MMP-dependent response. Nuclei from [GFP]LMNA-overexpressing cells slowed down or even stalled within the 2.5 μm-diameter constrictions of microfabricated channels as compared to [GFP]H2B-expressing cell nuclei that crossed the constriction in 1–2 h (Supplementary Fig. 2a–d). These findings are in agreement with increased nuclear stiffness induced by elevated LMNA levels[1,5]. [GFP]LMNA overexpression did not affect MT1-MMP level nor association of LINC complex components Nesprin-1 and SUN1 to the NE (Supplementary Fig. 2a and e, f). Invasion speed of MDA-MB-231 cells in 20 °C polymerized collagen gel (large pore size) was not affected by overexpression of [GFP]LMNA as compared to [GFP]H2B (Fig. 1h). However, contrasting with the reduction of collagenolysis by MDA-MB-231 cells in the permissive large pore size collagen gel

(Fig. 1f), collagenolysis levels of $^{GFP}$LMNA-overexpressing cells remained elevated in large, as compared to small, pore size collagen environment (Fig. 1i and Supplementary Fig. 2g). These observations revealed a relationship between LMNA overexpression, known to increase nuclear stiffness and elevated

levels of MT1-MMP-mediated ECM proteolysis during 3D invasion.

Reciprocally, reduction of LMNA levels has been shown to increase nuclear deformability[4,22,23]. LMNA was silenced to ≤5% of endogenous levels using two independent siRNAs with no

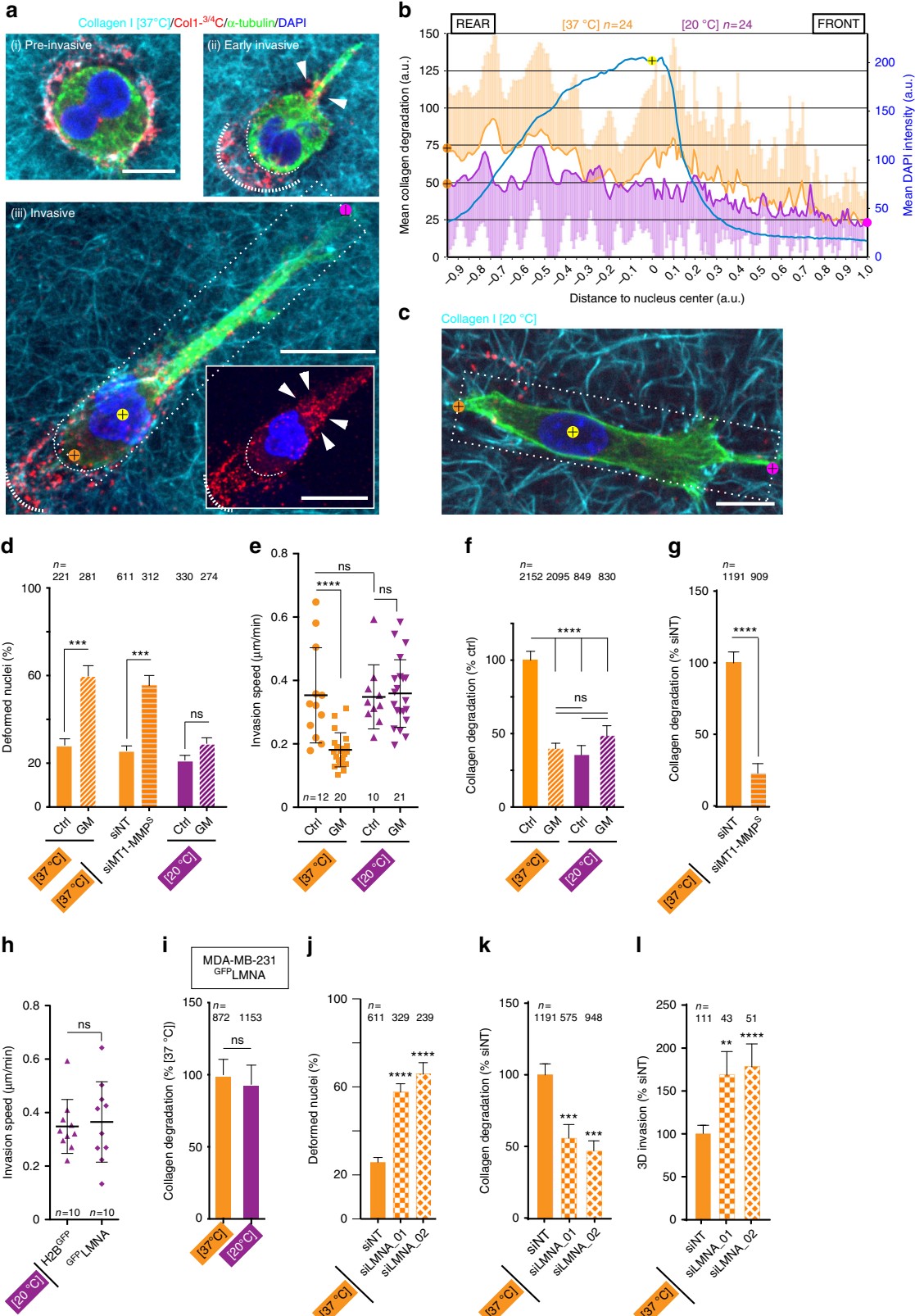

change in MT1-MMP expression nor visible alteration in cytoskeletal organization (Supplementary Fig. 3a, b). Reduced LMNA levels correlated with increased nuclear deformation in 3D collagen gel polymerized at 37 °C (55–65% deformed nuclei including 25% polymorphic lobulated nuclei, Fig. 1j) and in cells plated on a thick fibrous collagen layer (Supplementary Fig. 3b). In addition, LMNA KD correlated with increased migration speed of [GFP]H2B-expressing cells in microchannels (Supplementary Fig. 2c, d). Collectively these data indicated that LMNA-depleted cells had more deformable nuclei and retained full migratory capacity. Remarkably, MT1-MMP-dependent collagenolytic activity in the small pore size collagen gel decreased by ~2-fold upon LMNA KD (Fig. 1k and Supplementary Fig. 3c). Despite reduced collagenolysis, silencing of LMNA correlated with a 1.5–1.7-fold increase of invasion (Fig. 1l and Supplementary Fig. 3d). Thus, reduced LMNA expression enhanced the invasive potential of breast tumor cells by increasing nucleus deformability and its capacity to squeeze through constricted spaces, while MT1-MMP response was lowered down. All together, these findings indicated that MT1-MMP-mediated pericellular collagenolysis is an adaptive response that is switched on during confined migration in the dense ECM environment, suggesting a relationship between environment, nuclear biomechanics and modulation of active MT1-MMP at the cell surface.

**Polarization of MT1-MMP endosomes during confined migration**. In order to identify the mechanism underlying increased collagenolysis during constricted migration, we looked at the dynamic distribution of MT1-MMP-positive storage endosomes in MDA-MB-231 cells invading in 3D collagen by automated endosome tracking over time. In cells invading through the small pore size collagen gel, MT1-MMP-containing endosomes clustered in a region anterior to the nucleus relative to the direction of movement (Fig. 2a and Supplementary Movie 1);

~50% of MT1-MMP-positive endosomes polarized in a 90° quadrant in front of the nucleus relative to movement with a non-uniform distribution as demonstrated by Rao's Spacing test ($P < 0.001$, Fig. 2b). In contrast, we observed a striking uniform endosome distribution in MDA-MB-231 cells invading in the large pore size collagen gel polymerized at 20 °C ($P > 0.1$, Fig. 2d, e and Supplementary Movie 2). Relationship between MT1-MMP endosome polarity and collagen gel porosity was similarly observed in HT-1080 cells (Fig. 2c, f). Then, we examined the consequences of modulating LMNA levels, nuclear deformability and gel porosity on endosome polarization in MDA-MB-231 cells. In the permissive gel polymerized at 20 °C, decreased nuclear deformability upon LMNA[GFP] overexpression correlated with a remarkable polarization of MT1-MMP endosomes in front of the nucleus (Fig. 2g, h, compare to Fig. 2d, e, and see Supplementary Movie 3). Conversely, silencing of LMNA correlated with a loss of endosome polarization in front of the nucleus in cells invading through the confining gel polymerized at 37 °C and instead a bimodal endosome distribution was observed (Fig. 2i, j, compare to 2a, b, and see Supplementary Movie 4). Collectively, these findings suggested that endosome polarization in front of the nucleus is part of the response to adjust pericellular collagenolysis and widen constricting pores in the matrix.

**The centrosome is positioned ahead of the nucleus**. We observed that the centrosome was located in front of the nucleus in cells invading through the 3D small pore size collagen gel and noticed that the centrosome was often located in the vicinity of maximal nuclear constriction when invasion was impaired upon MMP inhibition (Fig. 3a, b). MT1-MMP endosome positioning and clustering around the centrosome are controlled by interactions with the microtubule network and dynein and kinesin motor activity[24]. Thus, nucleus-centrosome linkage and

---

**Fig. 1** MT1-MMP-dependent pericellular collagenolysis is an adaptive response to matrix porosity. **a** MDA-MB-231 cells were embedded in 2.0 mg/ml fluorescent type I collagen (cyan) and polymerization was induced at neutral pH at 37 °C. Cells were fixed after 2.5 hrs and stained for cleaved collagen neoepitope (Col1-³/⁴C antibody, red), α-tubulin (green) and DAPI (blue). Dashed box indicates the region used for line-scan analysis in panel b with yellow, orange and pink dots indicating nucleus center, posterior and anterior limits of regions used for line-scan analysis. Dashed lines indicate initial cell position (thick) and cell rear (thin), respectively. Inset shows nucleus and cleaved collagen signal and arrowheads point to nucleo-anterior collagenolysis. **b** Averaged maximal fluorescence intensity profiles of cleaved collagen in 37 °C (orange curve) or 20 °C (purple curve) polymerized collagen±SD (left Y-axis) and DAPI (right Y-axis) along cell axis. n, number of cells used to calculate averaged intensity profiles from three independent experiments; "0" on X-axis corresponds to nucleus center. **c** MDA-MB-231 cells embedded in 20 °C polymerized gel analyzed as in **a**. **d** Morphological analysis of DAPI-stained nuclei (see Supplementary Fig. 1a for nucleus shape scoring criteria) in MDA-MB-231 cells in 3D collagen matrix under indicated experimental conditions. Data are mean % ± SEM from three independent experiments (except Ctrl at [37 °C], N = 2 and siNT, N = 6); (n), number of cells analyzed. P-values of Kruskal–Wallis test as compared to control condition in each dataset. **e** MDA-MB-231 cells expressing H2B[GFP] were embedded in 3D 37 °C or 20 °C polymerized gels. Cells were treated with ethanol (Ctrl) or GM as indicated. Nuclei were automatically tracked from time-lapse sequences obtained from three independent experiments and plot shows the distribution of nuclei speed. n, number of cells analyzed from three independent experiments. Data were transformed using the log transformation $y = log(y)$ to make data conform to normality and analyzed using one-way ANOVA test. **f, g** Pericellular collagenolysis by MDA-MB-231 cells treated with GM (**f**) or silenced for MT1-MMP (**g**) in 37 °C or 20 °C polymerized gels measured as mean intensity of Col1-³/⁴C signal per cell (see Supplementary Fig. 1b for representative images). Values for vehicle-treated (panel f) or siNT-treated cells in 37 °C polymerized gel (**g**) were set to 100%. n, number of cells analyzed from three to five independent experiments (except experiments in 20 °C polymerized gel, N = 2); Kruskal–Wallis (**f**) and Mann-Whitney (**g**) tests. **h** MDA-MB-231 cells expressing H2B[GFP] or [GFP]LMNA were embedded in 3D collagen gel polymerized at 20 °C and invasion speed was analyzed as in **e**. n number of cells analyzed from three independent experiments. Unpaired t-test. **i** Pericellular collagenolysis by MDA-MB-231 cells expressing [GFP]LMNA in 37 °C (set to 100%) or 20 °C polymerized gels (see Supplementary Fig. 2g for representative images). n, number of cells analyzed from three independent experiments; Mann-Whitney test. **j** Analysis of nuclear deformation in MDA-MB-231 cells knocked down for LMNA in 37 °C polymerized gel as in **d**. Data are mean % ± SEM from three independent experiments (except siNT, N = 6); (n), number of cells analyzed. P-values of One-way ANOVA test as compared to control condition. **k** Pericellular collagenolysis by MDA-MB-231 cells knocked down for LMNA in 37 °C polymerized gel normalized to mean intensity of siNT-treated cells ± SEM (see Supplementary Fig. 3c for representative images); n number of cells analyzed from three independent experiments; Kruskal–Wallis test. **l** Relative invasion of cells penetrating 3D collagen to depths ≥ 30 μm (see Supplementary Fig. 3d for representative images). Data represent mean ± SEM normalized to invasion of control cells from three independent experiments. n number of cells analyzed from three independent experiments; Kruskal-Wallis test. **P < 0.01; ***P < 0.001; ****P < 0.0001; ns, not significant. Scale bar=10 μm

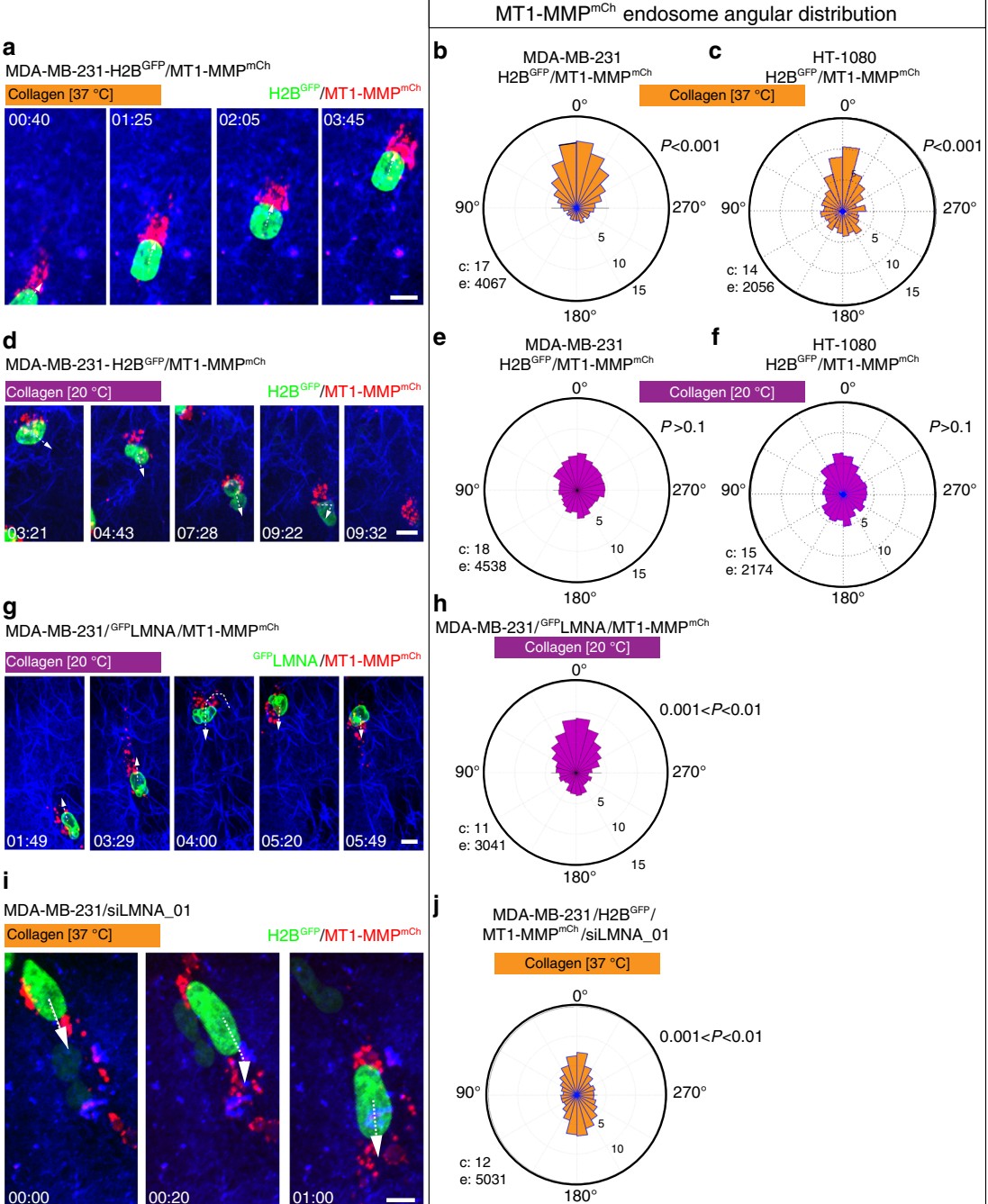

**Fig. 2** Polarization of MT1-MMP endosomes is part of the adaptive MT1-MMP-dependent collagenolysis response during confined invasion. **a** MDA-MB-231 cells expressing MT1-MMP$^{mCh}$ (red) and H2B$^{GFP}$ (green) were embedded in the 3D small pore size collagen gel and analyzed by time-lapse confocal spinning-disk microscopy (see Supplementary Movie 1). The gallery shows representative non-consecutive frames from a representative movie obtained from three independent experiments (time in h:min). Arrows show the direction of migration. **b**, **c** Rose plots showing the percentage of MT1-MMP endosomes in 15° segments relative to the direction of nucleus movement (0°) scored from time-lapse sequences of MDA-MB-231 (**b**) or HT-1080 (**c**) cells. c number of cells, e number of endosomes analyzed from three independent experiments. *P*-values for circular uniformity Rao's Spacing test are provided. **d** Representative frames from a time-lapse sequence of MDA-MB-231 cells invading in the large pore size collagen gel (see Supplementary Movie 2). **e**, **f** Angular distribution of MT1-MMP endosomes relative to the direction of nuclear movement as in **b**, **c**. **g** Gallery from a representative time-lapse sequence of MDA-MB-231 cells expressing MT1-MMPmCh (red) and $^{GFP}$LMNA (green) embedded in the large pore size gel (see Supplementary Movie 3). **h** Rose plot of MT1-MMP endosome angular distribution from three independent experiments. **i** Representative frames of a movie of MDA-MB-231 cells treated with siRNA against LMNA invading through 37 °C polymerized gel (see time-lapse in Supplementary Movie 4). **j** Rose plot of MT1-MMP endosome angular distribution from three independent experiments. Scale bars=10 μm

centrosome positioning in front of the nucleus could underlie polarization of MT1-MMP endosomes during constricted migration. Anchoring of the nucleus to the centrosome involves LINC complex components SUN and nesprins, which interact with cytoskeletal elements including microtubules[25–28]. We used overexpression of the dominant negative KASH domain of nesprin-2 ($^{GFP}$DN-KASH) known to antagonize SUN-nesprin interactions, which displaced nesprin-1 and -2 from the NE of

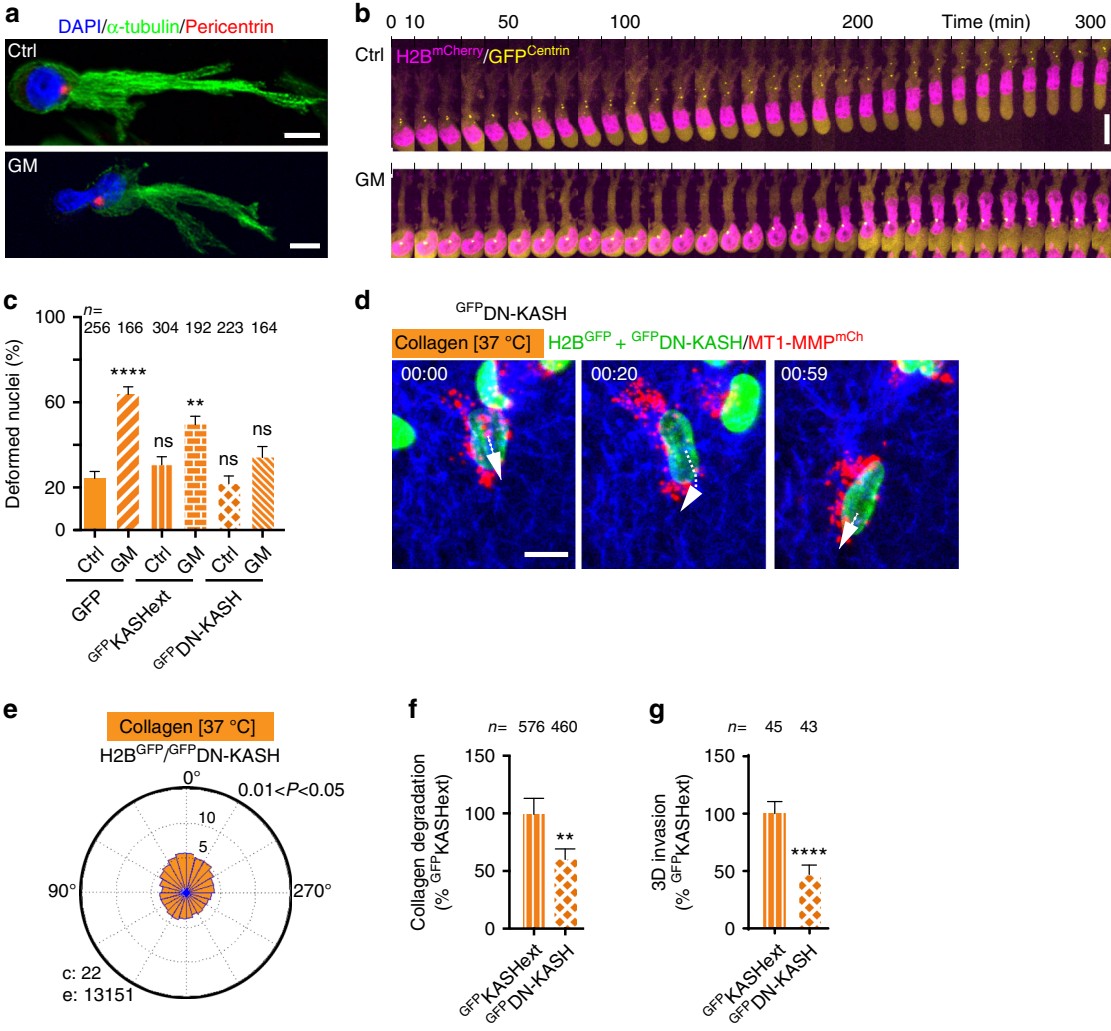

**Fig. 3** Nucleus-centrosome linkage through LINC complex is involved in MT1-MMP endosome polarization and collagenolysis-based invasion. **a** MDA-MB-231 cells in 3D collagen I polymerized at 37 °C treated or not with GM MMP inhibitor and stained for α-tubulin (green), centrosomal pericentrin (red) and nucleus (blue). **b** Galleries from representative time-lapse sequences of MDA-MB-231 cells expressing H2B^mCherry (magenta) and ^GFPCentrin-1 (yellow) invading through type I collagen gel as in **a**. **c** Morphological analysis of DAPI-stained nuclei as in Fig. 1d. Data are mean % ± SEM from three independent experiments. (*n*), number of cells analyzed. *P*-values of Kruskal–Wallis test as compared to non-treated GFP-expressing cells. **d** Representative frames of a movie of MDA-MB-231 cells expressing MT1-MMPmCh (red) and H2B^GFP together with ^GFPDN-KASH (green) invading through the small pore size collagen gel polymerized at 37 °C (see time-lapse sequence in Supplementary Movie 5). **e** Rose plots of endosome angular distribution from three independent experiments as in Fig. 2b. *P*-value for circular uniformity Rao's Spacing test is provided. **f** Pericellular collagenolysis by MDA-MB-231 cells expressing ^GFPDN-KASH in 37 °C polymerized gel normalized to mean intensity of ^GFPKASHext-expressing cells ± SEM (see Supplementary Fig. 4e for representative images); *n*, number of cells analyzed from three independent experiments; Mann–Whitney test. **g** 3D invasion of ^GFPDN-KASH-expressing cells in the small pore size gel normalized to invasion of ^GFPKASHext-expressing cells ± SEM from three independent experiments as in Fig. 1l. *n* number of cells analyzed from three independent experiments (see Supplementary Fig. 4f for representative images); Mann–Whitney test. **P < 0.01; ****P < 0.0001; ns non significant. Scale bars=10 μm

MDA-MB-231 cells (Supplementary Fig. 4a–c), and correlated with a ~10-fold increase in centrosome-nucleus distance (Supplementary Fig. 4d)[29,30]. In contrast, ^GFPKASHext with a C-terminal extension preventing binding to SUN did not affect nesprin association to the NE nor nucleus-centrosome linkage (Supplementary Fig. 4a–d). Increase in nuclear deformation during 3D invasion in the 37 °C polymerized gel upon MT1-MMP inhibition was suppressed when nucleus-centrosome linkage was loosened by ^GFPDN-KASH expression but not by ^GFPKASHext (Fig. 3c). Moreover, front polarization of MT1-MMP-positive endosomes was lost upon ^GFPDN-KASH overexpression in MDA-MB-231 cells in the small pore size collagen gel (Fig. 3d, e and Supplementary Movie 5). These effects were accompanied by a ~50% reduction of pericellular collagenolysis

and invasive potential upon ^GFPDN-KASH overexpression as compared to ^GFPKASHext (Fig. 3f, g and Supplementary Fig. 4e, f). To rule out some global cytoskeletal defects due to perturbation of nucleus-cytoskeletal linkage induced upon DN-KASH expression, we compared cytoskeletal changes induced by constitutively active Rac1 in cells expressing ^GFPDN-KASH or not. Expression of activated ^MycRac1L61 in MDA-MB-231 cells induced the archetypical cortactin-positive lamellipodial extension and cell spreading phenotype (Supplementary Fig. 4g, h). DN-KASH, like the inactive KASHext construct, did not interfere with Rac1L61-induced effects (Supplementary Fig. 4g, h). Thus we concluded that interfering with LINC complex function did not prevent actin filament assembly induced by Rac1 activation at the cell cortex. All together, these findings suggested that

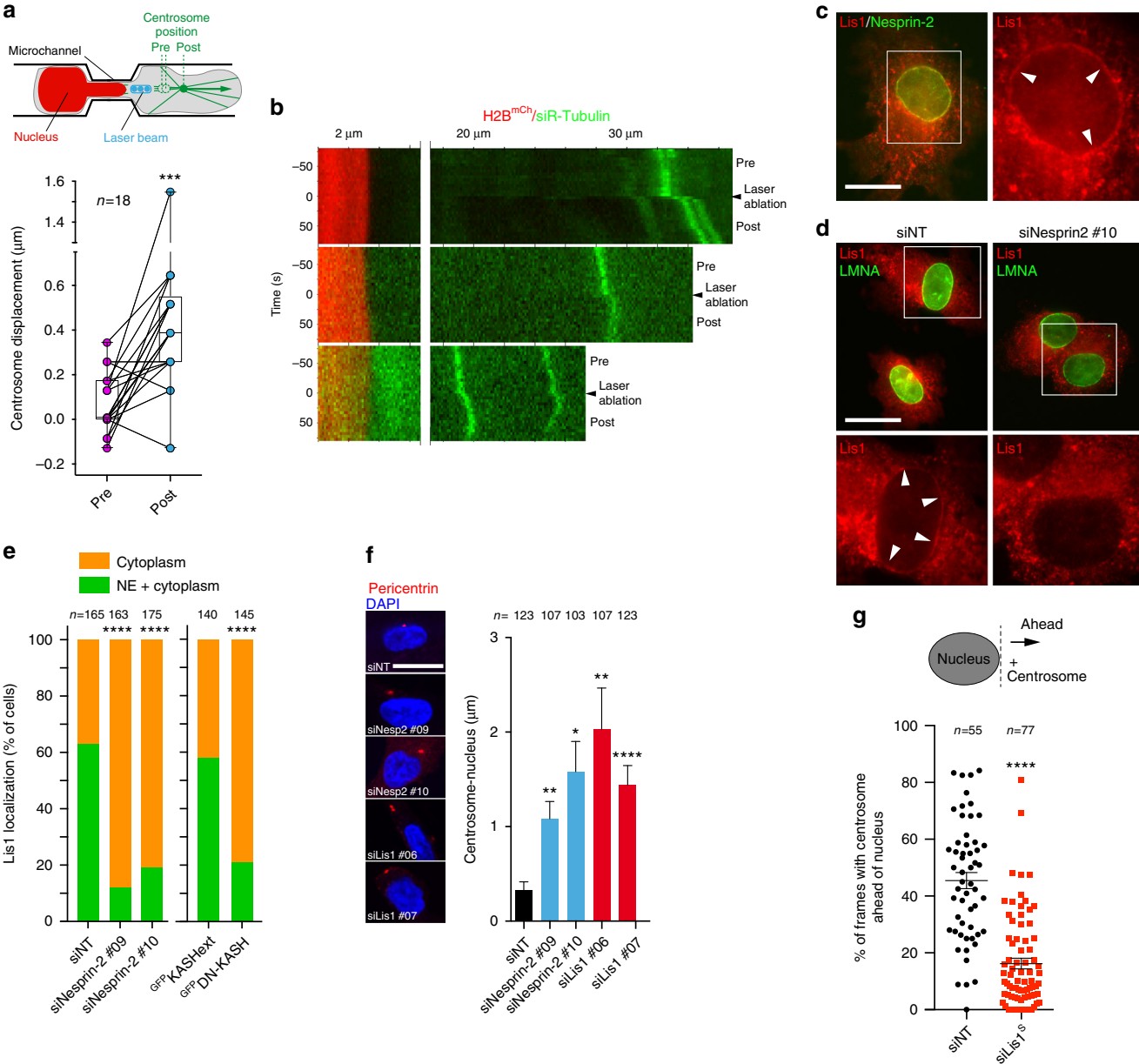

**Fig. 4** Lis1 contributes to nucleus-centrosome linkage in MDA-MB-231 cells. **a** Schematic representation of MDA-MB-231 cell expressing H2B[mCherry] and with SiR-tubulin-labeled microtubules migrating through the microfabricated channel. Centrosome displacement was measured before (pre) and after (post) irradiation with the laser beam (blue dots) focused between the nucleus and the centrosome. Dashed green circles, positions of the centrosome before laser irradiation; green dot, centrosome position after irradiation. For each cell analyzed ($n = 18$), the graph shows paired mean centrosome displacements (in μm) before (pre) and after (post) irradiation; Wilcoxon signed rank test. **b** Kymographs showing elastic recoil of the centrosome after laser irradiation (time 0). H2B[mCherry]-labeled nucleus is shown in red, SiR-tubulin-labeled microtubules and the centrosome are in green. **c** Lis1 and Nesprin-2 immunostaining in MDA-MB-231 cells treated with nocodazole. **d** MDA-MB-231 cells treated or not with nesprin-2 siRNA were treated with nocodazole and immunostained for Lis1 and LMNA. Insets in c and d represent Lis1 signal of the boxed regions. Arrowheads point at regions of Lis1 association with the NE. **e** Percentage of cells with Lis1 association with the NE was scored in cells treated with indicated siRNA. n, number of cells analyzed from three independent experiments; Fisher exact test. **f** Pericentrin and DAPI immunostaining. Scale bars=5 μm. Mean centrosome-nucleus distance (μm) in MDA-MB-231 cells in 3D collagen under indicated conditions ± SEM; n, number of cells analyzed from three (siLis1 and siNesprin-2) or two (siLMNA) independent experiments; Kruskal–Wallis test. **g** Centrosome position related to the nucleus was scored as schematized from time-lapse sequences of MDA-MB-231 cells expressing [GFP]centrin-1 and H2B[mCherry] and treated with Lis1 siRNA or with control siNT during invasion through the small pore size collagen gel. The graph represents the percentage of frames with the centrosome ahead of the front edge of the nucleus for each cell analyzed and the corresponding Box-and-whisker plot. n, number of cells analyzed from three independent experiments; Mann--Whitney test. *$P < 0.05$; **$P < 0.01$; ***$P < 0.001$; ****$P < 0.0001$. Scale bars=10 μm

perturbation of LINC complex function and nucleus-centrosome linkage interfered with front polarization of MT1-MMP storage compartments and with pericellular collagenolysis during confined migration in 3D.

**Nesprin-2 and Lis1 mediate nucleo-centrosome attachment.** Current models in neuronal cells suggest that nucleus-centrosome linkage involves interaction of cytoplasmic dynein anchored at the NE with centrosome-anchored microtubules, while

counterbalancing forces are exerted on the centrosome through cortically anchored microtubules[31–33]. We used laser ablation to probe forces in the nucleus-centrosome axis during confined migration of MDA-MB-231 cells in microfabricated channels (Fig. 4a). Centrosome movement was recorded by time-lapse

imaging, revealing low-amplitude oscillatory movement (Fig. 4a, b, pre). Microtubules located between the nucleus and the centrosome were then ablated by UV laser irradiation; the centrosome underwent elastic recoil towards the cell leading edge probably due to unbalanced pulling forces exerted from cortical sites in the

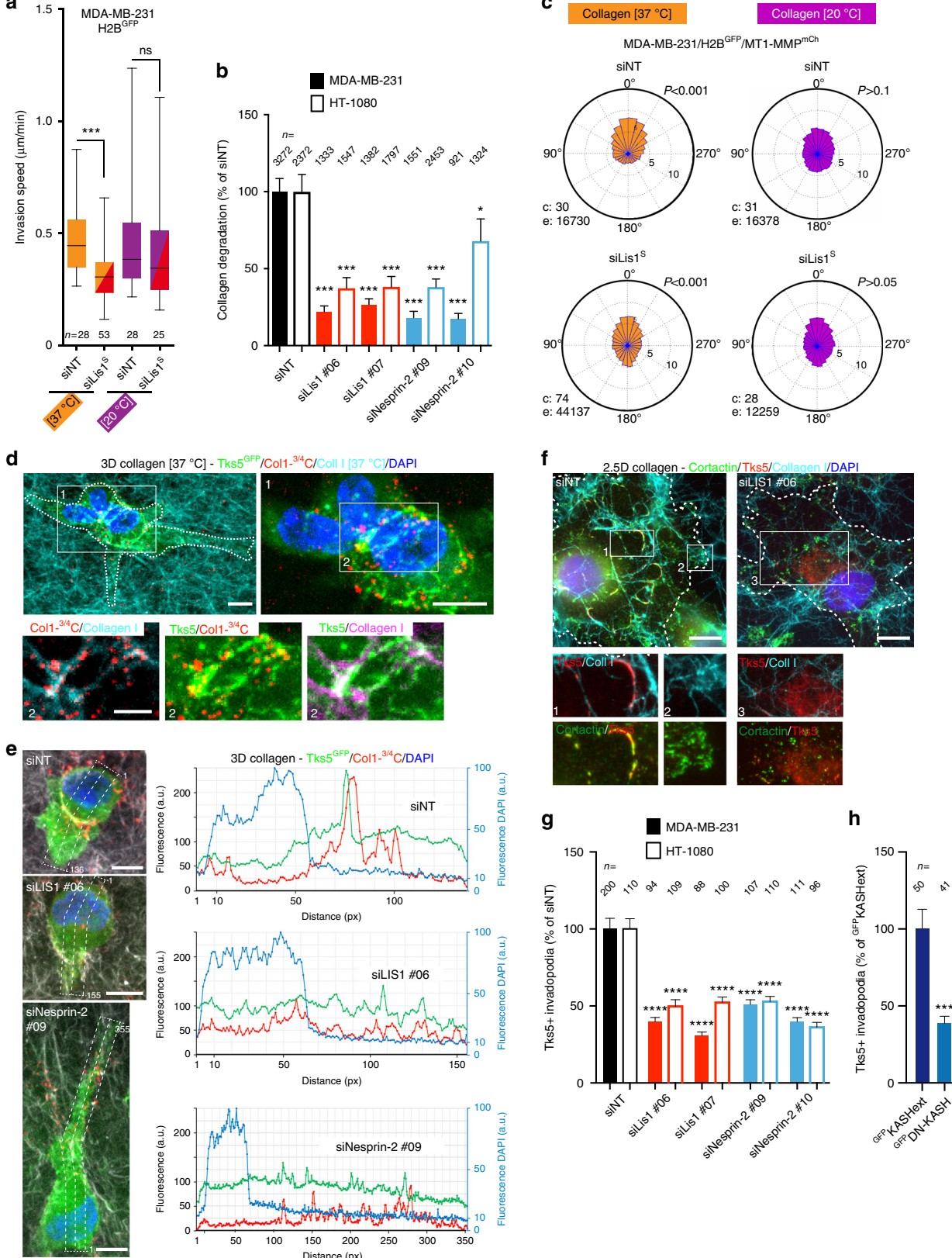

advancing cell protrusion (Fig. 4a, b, post). All together, these data supported the existence of tension forces in the nucleus-centrosome axis during confined migration of MDA-MB-231 cells.

Dynein motor and its regulator Lis1, which is essential for high-load dynein functions, have been implicated in nucleus-centrosome linkage during neuronal migration[34–37]. The prevailing model is that SUN-nesprin1/2 complexes mediate nucleo-centrosome attachment by providing anchors to cytoplasmic dynein/Lis1 complexes to pull the nucleus toward the centrosome[38]. Strong enrichment of Lis1 at the NE has been reported in different cell types upon nocodazole treatment[39–41]. Using similar conditions, we observed partial co-localization of Lis1 and Nesprin-2 at the NE in MDA-MB-231 cells (Fig. 4c). Lis1 was also associated with cytoplasmic vesicles (Fig. 4c, d). Interestingly, Nesprin-2 KD or [GFP]DN-KASH expression correlated with a significant reduction of Lis1 association with the NE (Fig. 4d, e). Collectively, these data suggested a contribution of nesprin-2 and LINC complex to Lis1 association with the NE. These observations also highlighted a possible function for Lis1 at the NE. Silencing of Lis1 in MDA-MB-231 cells (85–90% depletion by two independent siRNAs, Supplementary Fig. 5a, b) resulted in a significant 3–4-fold increase of centrosome-nucleus distance (Fig. 4f), similar to findings in Lis1-deficient neurons[19,34,35]. Silencing of Nesprin-2 (Supplementary Fig. 5c) also increased centrosome-nucleus distance (Fig. 4f), although to a lesser extent as compared to DN-KASH overexpression (Supplementary Fig. 4d). In addition, when centrosome position was scored relative to the nucleus based on movies of MDA-MB-231 cells invading through the small pore size collagen gel polymerized at 37 °C, we found that the centrosome was at ~50% of the time positioned ahead of the nucleus in control cells in agreement with data described above (Fig. 4g). Conversely, the centrosome was only at ~15% of the time positioned in front of the nucleus in cells knocked down for Lis1 (Fig. 4g). Based on these observations we concluded that interfering with Lis1 function affected centrosome positioning in front of the nucleus during confined migration in the collagen matrix, suggesting a role for Lis1 in nucleus-centrosome linkage in MDA-MB-231 cells.

**Nesprin-2 and Lis1 regulate nucleo-anterior collagenolysis.** The consequences of Lis1 silencing on MT1-MMP-based invasion were analyzed. Lis1 KD significantly reduced the invasion speed of MDA-MB-231 cells in the small pore size gel, while it did not affect invasion in the permissive 20 °C polymerized collagen environment (Fig. 5a). Correlating with the reduction of invasive potential in the small pore size collagen gel, depletion of Lis1 resulted in a strong reduction of pericellular collagenolysis by MDA-MB-231 and HT-1080 cells in the confining collagen environment similar to silencing of LINC complex component Nesprin-2 (Fig. 5b). In addition, we found that Lis1 depletion interfered with MT1-MMP endosome polarization ahead of the nucleus during confined migration in the small pore size collagen gel (Fig. 5c and Supplementary Movie 6). Endosome polarization was lost in the large pore size gel irrespective of Lis1 expression (Fig. 5c). Of note, silencing of Lis1 in MDA-MB-231 cells did not alter the overall distribution of MT1-MMP endosomes (Supplementary Fig. 5d). In contrast, overexpression of the p50[Glued]/dynamitin subunit known to disrupt dynactin complex function led to a dramatic redistribution of MT1-MMP endosomes to the cell periphery (Supplementary Fig. 5d)[24,42]. Therefore, we concluded that Lis1 was unlikely to play a significant role in microtubule-based traffic of MT1-MMP-containing endosomes and that it mainly contributed to MT1-MMP-dependent invasion through the regulation of nucleus/centrosome linkage and centrosome positioning.

Collagen degradation is mediated by F-actin, Tks5-positive invadopodia forming at the cell cortex in association with collagen fibrils, where MT1-MMP accumulates[43–47]. MDA-MB-231 cells embedded in a small pore size 3D collagen network formed typical Tks5-positive structures in association with nucleus-constricting fibrils and with collagenolysis (Fig. 5d, e, siNT). In contrast, cells knocked down for nesprin-2 or Lis1 showed reduced accumulation of Tks5 and collagen degradation in front of the nucleus (Fig. 5e), in agreement with global decrease in collagenolysis (Fig. 5b). Cortactin- and Tks5-positive structures were also visible at the ventral surface of MDA-MB-231 cells in contact with a ~5–10 μm thick (2.5D) layer of fibrous collagen polymerized at 37 °C allowing quantification of invadopodia formation (Fig. 5f)[46,47]. Silencing of Lis1 in MDA-MB-231 or HT-1080 cells resulted in a strong inhibition of the formation of Tks5-positive structures (Fig. 5f, g). Total level of Tks5 or MT1-MMP proteins were not affected by Lis1 silencing (Supplementary Fig. 5e). Endogenous Lis1 protein was silenced and Lis1 levels were restored by transfection with a cDNA encoding a siRNA-resistant variant of Lis1[GFP] (Lis1[R#06/GFP], Supplementary Fig. 5f, g). Lis1[R#06/GFP] rescued assembly of Tks5-positive structures in knocked down cells similar to control levels arguing for a specific effect of Lis1 depletion (Supplementary Fig. 5h). Silencing of Nesprin-2 similarly interfered with formation of Tks5 structures (Fig. 5g), as did disruption of LINC complex function by DN-KASH (Fig. 5h). Altogether, our findings indicated that interfering with LINC complex and Lis1 function in nucleus-centrosome attachment affects pericellular collagenoysis in association with cortical cell-matrix contact sites (see model in Fig. 6).

**Fig. 5** Lis1 and LINC complex function is required for invadopodia assembly and activity. **a** Effect of Lis1 KD on invasion speed of MDA-MB-231 cells expressing H2B[GFP] in 3D collagen gels polymerized at 37 °C or 20 °C (as in Fig. 1e). n, number of cells analyzed from three independent experiments. Data were transformed using the log transformation $y = \log(y)$ to make data conform to normality and analyzed using one-way ANOVA test. **b** Pericellular collagenolysis (mean Col1-$^{3/4}$C signal per cell normalized to mean intensity of siNT-treated cells ± SEM) by MDA-MB-231 and HT-1080 cells silenced for Lis1 or Nesprin-2 in collagen polymerized at 37 °C (as in Fig. 1k). n, number of cells analyzed from three independent experiments. Kruskal–Wallis test as compared to siNT. **c** Angular distribution of MT1-MMP[mCh] endosomes in Lis1-depleted cells in 37 °C or 20 °C polymerized collagen gel (as in Fig. 2b). See Supplementary Movie 6. **d** Maximal projection of a series of sixteen confocal sections (7.5 μm width) of MDA-MB-231 cells expressing Tks5[GFP] (green) in 3D collagen gel polymerized at 37 °C (Cyan) stained for cleaved collagen (red) and nucleus (blue). Scale bars, 10 μm. Bottom row, two-by-two channel combinations corresponding to boxed region #2. Scale bar=5 μm. **e** MDA-MB-231 cells silenced for Lis1 or nesprin-2 embedded in 3D collagen as in **d**, stained for Tks5[GFP] (green) and cleaved collagen (red). Right row shows intensity profiles of Tks5[GFP], Col1-$^{3/4}$C and DAPI maximum intensities along the long-cell axis (dotted line-enclosed regions). **f** MDA-MB-231 cells silenced for Lis1 were incubated on a thick fibrous type I collagen layer (cyan). Invadopodia in association with collagen fibrils are labeled for cortactin (green) and Tks5 (red). Insets, two-by-two channel combinations of boxed regions. Tks5 is excluded from cortactin-positive lamellipodia (inset #2). Scale bars=10 μm. **g** Quantification of Tks5 signal in MDA-MB-231 and HT-1080 cells treated with indicated siRNAs plated on a thick layer of type I collagen. Y-axis indicates ratio of Tks5 area to total cell area normalized to mean value in siNT-treated cells (as percentage) ± SEM. n, number of cells analyzed from three independent experiments. Kruskal–Wallis test. **h** Quantification of Tks5 signal in MDA-MB-231 expressing [GFP]KASHext or [GFP]DN-KASH as in panel g from two independent experiments; Mann–Whitney test. *P < 0.05; ***P < 0.001; ****P < 0.0001

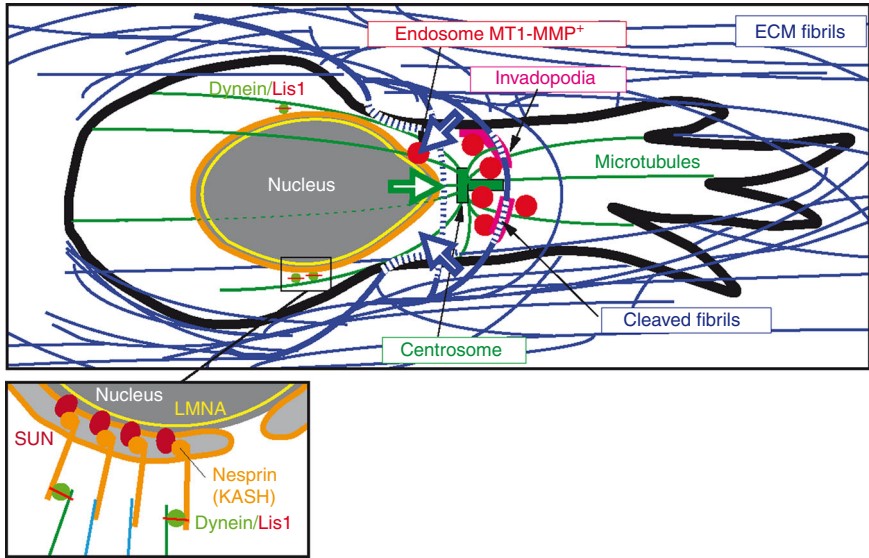

**Fig. 6** Model of LINC complex-, Lis1-dependent nucleus-centrosome linkage control of MT1-MMP matrix digest-on-demand response. Confined migration of tumor cells through dense 3D collagen network results in nucleus confinement by constricting collagen fibrils. Nucleus-microtubule cytoskeleton linkage through LINC complex and dynein heavy load factor Lis1 and cortical anchoring of microtubules is required for centrosome and MT1-MMP endosome positioning and for targeted delivery of MT1-MMP to invadopodia. Nucleus movement is facilitated by localized invadopodia-based pericellular proteolysis of confining fibrils ahead of the nucleus. Open arrows represent nuclear pulling force and counteracting forces from the matrix. Inset, scheme of nucleus-cytoskeletal linkage through LINC complex components nesprin and SUN in association with lamins. Lis1, probably in complex with dynein associates to the NE depending on Nesprin-2 and is involved in nucleus-microtubule linkage

## Discussion

Several converging studies revealed that nuclear stiffness and nuclear deformability are critical factors that limit confined cell migration through adjacent tissue and basement membrane transmigration by carcinoma cells[1–4,7]. In addition, recent reports have shown that confinement can generate mechanical stress on the nucleus as exemplified by nuclear deformations and loss of NE integrity and DNA damage[3,17,48,49]. Increased nucleus deformability as a consequence of LMNA down-modulation can facilitate the migration of cancer cells through small constrictions in reconstituted 3D matrix environments[2–4,8,9]. Nucleus migration through constricted spaces is also central to several developmental processes[50].

Our findings reveal a novel mechanism in which tumor cells adapt to the 3D matrix environment based on a digest-on-demand response to support confined migration in the tissue matrix. This mechanism depends on LINC complex-mediated linkage of the nucleus to the microtubule-centrosome network and to high-load dynein adaptor Lis1 (see Fig. 6). It is known that mechanical coupling of the nucleus to the cell cortex through cytoskeletal elements enables force transmission to the nucleus in a LINC complex-dependent manner[30,51]. Our data suggest a model whereby anchoring of dynein and its regulator Lis1 to the NE mediated by the LINC complex is involved in generation of forward nucleus pulling force on the microtubule-centrosome system[19,26,35,52–54]. We hypothesize that nucleus forward-pulling forces generate nuclear deformation due to the resistance opposed by constricting matrix fibrils during confined cell migration and that nucleo-cortex linkage may contribute to the assembly of invadopodial structures at plasma membrane-matrix contact sites resulting in focal degradation of confining fibrils (Fig. 6). It may seem paradoxical that interfering with LINC complex or Lis1 function affects invadopodia formation in cells cultured on top of a 2D collagen layer (see Fig. 5f–h). However, under these conditions we observed some pulling and pushing activities of cells on the surrounding collagen fibrils, and cells also squeezed through fibrils, all situations that may lead to some level of

physical constraints and confinement. Noticeably, a previous study reported no effect of SUN1 or nesprin-2 KD on invadopodia formation and function in a melanoma cell line[55]. This discrepancy with our data may be related to difference in matrix environment, i.e., 2D non-fibrillar denatured collagen (gelatin) vs. fibrous native type I collagen in our study with distinct collagenic receptors contributing to invadopodia formation[55–59].

Additional mechanisms of tumor cells have been described that lead to nuclear deformation through formation of perinuclear actin- and actomyosin-based structures in association with LINC complex components including nesprin-family proteins. These mechanisms enable nucleus squeezing through narrow spaces and correlate with increased invasiveness and metastatic potential[60–63]. In addition, a nuclear piston mechanism based on actomyosin contractility has been described, which generates high pressure within the anterior cytoplasmic compartment and enables migration through the 3D confining matrix[64]. This mechanism was recently reported to represent a strategy for nuclear forward movement through confining 3D environments that could compensate for low MMP activity[65]. The adaptive MT1-MMP-based collagenolysis response to confinement that we described here and the nuclear piston mechanism may represent different molecular systems consequential to nuclear stiffness and nuclear forward movement during confined migration. An intriguing possibility is whether the nuclear piston mechanism may contribute to nucleo-anterior polarization of MT1-MMP compartments. More work will be needed to determine the contribution and integration of these mechanisms to the metastatic program to tumor cell plasticity. Future studies should also unravel how these mechanisms are integrated with and possibly control invadopodia formation and targeted delivery of MT1-MMP to the cell surface for dissolution of confining ECM fibrils.

## Methods

**Plasmid constructs.** Construct expressing Tks5[GFP] was a kind gift of Dr S. Courtneidge (OHSU, Portland, OR). Retroviral vector encoding H2B[GFP] and [GFP]LMNA were provided by F. A. Dick (UWO, London, ON, Canada) and T.

Misteli (Addgene #17662), respectively. Lentiviral vector encoding H2B[mCh] was from M. Mercola (Addgene #21217). The [GFP]Centrin-1 plasmid was obtained from M. Bornens (Institut Curie, Paris, France). pRK5[myc]Rac1L61 was a kind gift of Dr S. Etienne-Manneville (Institut Pasteur, Paris, France). [GFP]DN-KASH and [GFP]KASHext were generated by inserting the C-terminal domain of Nesprin-2 (corresponding to the last 331 amino acids) into the XhoI–BamHI site of pECFP-C1 (Takara Bio Inc.). KASHext was derived from the DN-KASH construct by adding a C-terminal VDGTAGPGSTGSR amino acid extension[29]. For Lis1 rescue experiment, LIS1[GFP] construct was mutagenized to make it resistant to siLIS1#06 siRNA using the QuikChange Primer Design kit (Agilent) using: forward primer; 5′-CTCTGCTTCcGAGGATGCTACTATcAAaGTtTGGGAcTACGAGAC TGGAGATTTTGAA-3′; reverse primer: 5′-AAAATCTCCAGTCTCGTAgTCCC AaACtTTgATAGTAGCATCCTCgGAAGCAGAG-3′.

**Cell culture, stable and transient transfection and siRNA treatment**. MDA-MB-231 cells (ATCC HTB-26) were grown in L15 medium supplemented with 15% foetal calf serum and 2 mM glutamine at 37 °C in 1% CO2. HT-1080 fibro-sarcoma cells (ATCC CCL-121) were grown in DMEM GlutaMAX supplemented with 10% foetal calf serum. Cell lines were obtained from ATCC and were routinely tested for mycoplasma contamination. MDA-MB-231 cells stably expressing H2B[GFP/mCh], MT1-MMP[mCh] or [GFP]LMNA and MT1-MMP[mCh] or HT-1080 cells stably expressing MT1-MMP[mCh] were generated by lentiviral transduction. For transient expression, MDA-MB-231 cells or HT-1080 cells were transfected with plasmid constructs using AMAXA nucleofection (Lonza). Cells were analyzed by live cell imaging 24–48 h after transfection. For knockdown, MDA-MB-231 cells were treated with the indicated siRNA (50 nM, Dharmacon) using Lullaby (OZ Biosciences, France) and analyzed after 72 h of transfection. The following siRNAs were used: siNT (Non Targeting), siLMNA-01: 5′-GGUGGUGACGAUC UGGGCU-3′; siLMNA-02: 5′-CUGGACAGGUGGUGACGAU-3′; siMT1-MMP[S] (Smartpool): 5′-GGAUGGACACGGAGAAUUU-3′; 5′-GGAAACAAGUACUAC CGUU-3′; 5′-GGUCUCAAAUGGCAACAUA-3′; 5′-GAUCAAGGCCAAUGUU CGA-3′; siLis1[S] (Smartpool): 5′-CAAUUAAGGUGUGGGAUUA-3′ (siLis1#06); 5′-UGAACUAAAUCGAGCUAUA-3′ (siLis1#07); 5′-GGAGUGCCGUUGAUUG UGU-3′; 5′-UGACAAGACCCUACGCGUA-3′; siNesprin-2[S] (Smartpool): 5′AGG AAUUUCUGCAAACCGA-3′ (siNesprin-2#09); 5′GGUAGAACGUCAACCUCA A-3′ (siNesprin-2#10); 5′CCUAGAGUGUCGGAGGGAA-3′; 5′CACAGGAGCU UCACAAUAA-3′.

**Antibodies and reagents**. The source and working dilution of commercial anti-bodies used for this study are listed in Supplementary Table 1. Monoclonal anti-body against Nesprin-2A has been previously described[66,67]. Nocodazole (Sigma) was diluted in DMSO and used at a concentration of 10 μM. GM6001 (Millipore) diluted in ethanol was used at a concentration of 40 μM. Hepatocyte growth factor (HGF) was purchased from PeproTech Inc. and used at 20 ng/ml.

**Western blot analysis**. Cells were lysed in SDS sample buffer, separated by SDS-PAGE, and detected by immunoblotting analysis with the indicated anti-bodies. Antibodies were visualized using the ECL detection system (GE Healthcare)

**Indirect immunofluorescence microscopy**. Samples were fixed with 4% paraf-ormaldehyde, permeabilized with 0.1% Triton X-100, and then incubated with indicated antibodies. For alpha-tubulin staining samples were fixed with 4% par-aformaldehyde at 37 °C for 30 min. For better visualization of Lis1 association with the NE, cells were incubated for 1 h in nocodazole (10 μM) prior to fixation[35,39–41,68]. The analysis of MT1-MMPmCh endosome position relative to the cell center/cell periphery axis was performed, as described in ref. [24].

**Invadopodia formation assay**. Coverslips were layered with 200 μl of ice-cold 2.0 mg/ml acidic extracted collagen I solution (Corning) in 1 × MEM mixed with Alexa Fluor 647-conjugated type I collagen (5% final). The collagen solution was adjusted to pH7.5 using 0.34 N NaOH and Hepes was added to 25 μM final concentration. After 3 min of polymerization at 37 °C, the collagen layer was washed gently in PBS and 1 ml of the cell suspension in L15 medium with 15% FCS (10^5 cells/ml) was added. Cells were incubated for 90 min at 37 °C in 1% CO2 before fixation. Cells were pre-extracted with 0.5% Triton X-100 in 4% paraformaldehyde in PBS during 90 s and then fixed in 4% paraformaldehyde in PBS for 20 min and stained for immunofluorescence microscopy with Tks5 and Cortactin antibodies. Images were acquired with a wide-field microscope (Eclipse 90i Upright; Nikon) using a 100 × Plan Apo VC 1.4 oil objective and a highly sensitive cooled interlined charge-coupled device (CCD) camera (CoolSnap HQ2; Roper Scientific). A z-dimension series of images was taken every 0.2 μm by means of a piezoelectric motor (Physik Instrumente). For quantification of Tks5 associated with curvilinear invadopodia in cells plated on collagen fibers, five consecutive z-planes corresponding to the plasma membrane in contact with collagen fibers were projected and surface covered by Tks5 signal was determined using the thresholding command of ImageJ excluding regions <8 pixels to avoid non-invadopodial structures. Surface covered by Tks5 was normalized to the total cell surface and values normalized to control cells.

**Quantification of pericellular collagenolysis**. Cells treated with indicated siRNAs and expressing indicated GFP-tagged constructs were trypsinized and resuspended (2.5 × 10^5 cells/ml) in 0.2 ml of ice-cold 2.0 mg/ml acidic extracted collagen I solution in 1 × MEM, pH 7.5 buffer. The pH of the collagen solution was raised to 7.5 using 0.34 N NaOH and Hepes was added to 25 μM final concentration. 40 μl of the cell suspension in collagen was added on a 18 mm-diameter glass coverslip and collagen polymerization was induced by incubation for 90 min at 20 °C or 30 min at 37 °C. After polymerization, complete medium was added and collagen-embedded cells were incubated for 16 h at 37 °C. After fixation in 4% paraformaldehyde in PBS at 37 °C for 30 min, samples were incubated with anti-Col1-^3/4^C antibodies (2.5 μg/ml) for 2 h at 4 °C, washed extensively with PBS and counterstained with Cy3-conjugated anti-rabbit IgG antibodies and with Phalloïdin-Alexa488 to visualize cell shape and with DAPI. Image acquisition was performed with an A1r Nikon confocal microscope with a 40 × NA 1.3 oil objective using high-sensitivity GaASP PMT detector and a 595 ± 50 nm band-pass filter. Quantification of degradation spots was performed as previously described[47]. Briefly, maximal projection of 10 optical sections with 2 μm interval from confocal microscope z-stacks (20 μm depth) were preprocessed by a laplacian of Gaussian filter using a homemade ImageJ macro (available as sup-plementary information in[47]). Detected spots were then counted and saved for visual verification. No manual correction was done. Degradation index was the number of degradation spots divided by the number of cells present in the field, normalized to the degradation index of control cells set to 100. Nuclear defor-mation was visually and qualitatively assessed from maximal projection of 10 optical sections of DAPI signal from confocal microscope z-stacks (20 μm depth) by scoring nuclei as "normal" or "deformed" using criteria as described in Sup-plementary Fig. 1a.

**3D type I collagen invasion assay**. 200 μl of 2.0 mg/ml Collagen I was allowed to polymerize in transwell inserts (Corning) for 2 h at 37 °C as above. Cells were seeded on top of the collagen gel in complete medium and 20 ng/ml HGF was added to the medium in the bottom chamber of the transwell as chemoattractant. After 48 h of seeding, cells were fixed and stained with DAPI and visualized by confocal microscopy with serial optical sections captured at 10-μm intervals with a ×10 objective on a Zeiss LSM510 confocal microscope. Invasion was measured by dividing the sum of DAPI signal intensity of all slides beyond 30 μm (invading cells) by the sum of the intensity of all slides (total cells).

**Live-cell imaging in 3D type I collagen**. For Inter-fibril distance estimation, a 50 μl drop of 2.0 mg/ml fluorescently-labeled Collagen I was allowed to polymerize for 2 h at 37 °C or 20 °C as described above. Distances between collagen fibrils were measured from stacks of 30 optical sections acquired at 0.5 μm-interval with a SP8 Leica laser confocal microscope in the xy, xz, and yz planes (15 μm depth) using a 63 × 1.4NA oil objective, 4 detection channels (2 PMTs and 2 hybrid Detectors) and 405, 488, 561 and 633 nm laser lines. The system was steered by Leica Application Suite (LAS-X) software. For live cell imaging, glass bottom dishes (MatTek Corporation) were layered with 10 μl of a solution of 5 mg/ml unlabeled type I collagen mixed with 1/20-40 volume of Alexa Fluor 647-labeled collagen. Polymerization was induced at 37 °C or 20 °C for 3 min as described above, and the bottom collagen layer was washed gently in PBS and 1 ml of cell suspension (1.5–2.5 × 10^5 cells/ml) in complete medium was added. Cultures were incubated for 30 min at 37 °C, then medium was gently removed and two drops of a mix of Alexa Fluor 647-labeled type I collagen/unlabeled type I collagen at 2.0 mg/ml concentration were added on top of the cells (top layer). After polymerization at 37 °C or 20 °C for 90 min as described above, 1 ml of medium containing 20 ng/ml HGF was added to the cultures. z-staks of images were acquired every 5 min (150 ms exposure time) during 16 h by confocal spinning disk microscopy (Roper Scientific) using a CSU22 Yokogawa head mounted on the lateral port of an inverted TE-2000U Nikon microscope equipped with a 40 × 1.4NA Plan-Apo objective lens and a dual-output laser launch, which included 491 nm and 561 nm 50 mW DPSS lasers (Roper Scientific). Images were acquired with a CoolSNAP HQ2 CCD camera (Roper Scientific). The system was steered by Metamorph 7 software.

**Automated tracking of endosome angular distribution**. A homemade Matlab program (available on demand) was developed to track nuclei based on nuclear staining and create a velocity-dependent coordinate system to analyze MT1-MMP endosomes relative to the direction of displacement of the nucleus. Nuclei were automatically segmented from maximal z-stack projection of sequential time frames (see previous section) based on smoothing and thresholding and then were tracked based on the distance from their previous position. From the trajectory of each nucleus, speed (μm/min) and directionality (persistence) were computed for each consecutive pair of frames. A new polar coordinate system was defined such that the gravity center of the nucleus becomes the position (0,0) for all time points and that the velocity direction had an angle of 0°. This coordinate system was then changing for each time point and was different for each nucleus. Endosomes around each nucleus were automatically segmented by laplacian of gaussian spot enhancement and marker-control watershed segmentation based on regional maxima. The coordinates of the positions of the center of gravity of all endosomes

were then converted to this nucleus velocity-dependent coordinate system. Endosomes exactly in front of the nucleus in the direction of movement are then at 0° and endosomes exactly at the rear of the displacement vector are at 180°. All data created for endosomes for all processed nuclei and all movies for one condition were then pooled to create a polar histogram (radar plot), showing the distribution of endosomes relative to the direction of nuclear movement.

**Averaged Col1-3/4C intensity profiles**. Col1-3/4C neoepitope (collagen degradation) and DAPI (nucleus) intensity profiles were obtained using the line-scan function (maximal intensity) of Metamorph analyzing the back-to-front cell region including the nucleus for both signals (see Fig. 1a, c). Then, we used a homemade Matlab program (available on demand) to normalize line-scans to correct for line-scan length difference. Briefly, three reference points were manually defined based on the DAPI signal profile; the nucleus center and the back and front of the cell, respectively. These reference points were used to define a linear transformation such that they became 0, −1, and 1 coordinates, respectively on the normalized curves, allowing direct comparison of different profiles.

**Centrosome-nucleus distance measurement**. Cells were embedded in 2.0 mg/ml type I collagen polymerized at 37 °C as above. After 16 h, cultures were fixed with 4% paraformaldehyde and stained using polyclonal rabbit anti-pericentrin antibodies. Detection was performed with fluorescently-labeled anti-rabbit antibody. DNA was stained with DAPI. Centrosome-to-nucleus distance was determined by overlaying pericentrin and DAPI images and extending a line from the centrosome (center of pericentrin staining) to the nearest point of the nucleus rim; length of this line was measured using ImageJ software tools.

**Microfabrication of microchannels**. Micro-channels were prepared as previously described[69]. Briefly, polydimethylsiloxane (PDMS) (GE Silicones, 10/1 w/w PDMS A/crosslinker B) was used to prepare 7 μm-wide micro-channels with 2.5 μm constrictions from a self-made mold. Channels with constrictions were washed with PBS at least three times and incubated with complete medium for at least 5 h before adding the cells.

**Laser ablation**. MDA-MB-231 cells expressing H2B[mCh] migrating in PDMS channels were labeled with 50 mM SiR-tubulin-Cy5 (Spirochrome) in medium containing 20 ng/ml HGF during 3 hrs at 37 °C. Cells with the nucleus passing through the constrictions were selected. Z stacks (4 images, 0.5 μm z-step) images were acquired at 5 s interval during 75 s (pre-ablation). For photoablation, the 355 nm laser beam was focused to a region of interest selected between the nucleus and SiR-tubulin-labeled centrosome during a 40–80 ms pulse at 50–80% laser power. The conditions of ablation were monitored by the absence of recovery of SiR-tubulin signal. Z stacks were acquired as above for 25 s (post-ablation). The centrosome displacement was measured before and after laser ablation over 25 s periods of time and scored as positive in the direction of cell movement or negative otherwise. Pre- and post-ablation displacements are measured in the same cell and paired in the statistical analysis.

**Statistics and reproducibility**. All results are presented as the mean ± SEM of three independent experiments except for collagen degradation in 20 °C polymerized gel Fig. 1f ($N = 2$), centrosome-nucleus distance analysis in siLMNA-treated cells in Fig. 3e ($N = 2$) and Tks5 recruitment analysis in Fig. 5h ($N = 2$). GraphPad Prism (GraphPad Software) was used for statistical analysis. Statistical significance was defined as $*P < 0.05$; $**P < 0.01$; $***P < 0.001$; $****P < 0.0001$; ns, not significant. Data were tested for normal distribution using the D'Agostino-Pearson normality test and nonparametric tests were applied otherwise. One-way ANOVA, Kruskal–Wallis, Mann–Whitney or Wilcoxon signed rank tests were applied as indicated in the figure legends. Radial distributions of endosome localization with respect to instantaneous direction of nuclear movement were plotted and analyzed using the "circular" R package[70,71]. Circular uniformity Rao's Spacing test was employed to test if angle distributions were significantly different from a uniform distribution (significant difference between data and a uniform distribution when $P < 0.05$).

**Data availability**. All data are available within the Article and Supplementary Files, or available from the authors upon request.

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

## Acknowledgements

We thank the Nikon Imaging Centre @ Institut Curie-CNRS and Cell and Tissue Imaging Facility of Institut Curie, member of the France Bio Imaging national research infrastructure (ANR-10-INBS-04) for help with image acquisition, Drs M. Bornens, S. Courtneidge, S. Etienne-Manneville, A-M Lennon-Duménil, F. Perez, M. Mercola, T. Mistelli, Q. Zhang and R. Vallee for providing reagents for this study and Dr I. Brito for help with statistical analysis. E.I. was supported by a postdoctoral fellowship from Ligue Nationale contre le Cancer, P.M. by a fellowship from Fondation ARC pour la Recherche contre le Cancer, R.F. by a fellowship from Ministère de l'Education Nationale, de l'Enseignement supérieur et de la Recherche, A.C. by a grant from Worldwide Cancer Research (Grant 16-1235 to P.C.) and S.A.G. by a grant provided by the program «Investissements d'Avenir» launched by the French Government and implemented by Agence Nationale pour la Recherche (ANR) with the reference ANR-10-LBX-0038 (Labex CelTisPhyBio) part of the IDEX PSL (ANR-10-IDEX-0001-02 PSL). Funding for this work was provided by the program «Investissements d'Avenir» launched by the French Government and implemented by Agence Nationale pour la Recherche (ANR) with the reference ANR-10-LBX-0038 (Labex CelTisPhyBio) part of the IDEX PSL (ANR-10-IDEX-0001-02 PSL) and by grants from Ligue Nationale contre le Cancer (Equipe labellisée 2015) and from Worldwide Cancer Research (Grant 16-1235) to P.C. and by core funding from Institut Curie and Centre National pour la Recherche Sci-entifique (CNRS).

## Author contributions

E.I, A.C., R.F., and P.M were equally involved in experimental design, experiments and data analysis; S.A.G. designed and performed microfabricated channel experiments with help from M.R. and M.P.; M.J.D. performed experiments and analyzed data; P.P.G. generated software for automated tracking analysis and contributed to data analysis; Pa.M. performed circular statisticar tests; E.R.G. and A.B. suggested and contributed to designing experiments based on nucleus-cytoskeleton coupling; C.M.S. provided critical reagents and P.C. and E.I. conceived the study and wrote the manuscript.

## Additional information

**Competing interests:** The authors declare no competing interests.

