## [Peer Review File · Nature Communications]

Reviewers' comments:

Reviewer #1 (Remarks to the Author):

In the manuscript Infante et al the authors describe how proteolysis in substrate confinement is organized anterior to the bulky nucleus that -like a piston (now an established term by previous publications by Petrie & Yamada)- localizes MT1-MMP positive endosomes to its front, in order to have constricting substrate fibers degraded. This 'piston' mechanism only works in confinement and/ or if the nucleus is mechanically stiffened which the authors deduct from high versus low lamin A/C expression levels, as well as by an intact LINC-dynein-tubulin connection. The concept as such is very interesting, original and plausible and derives smoothly from earlier concepts in the field. If well presented, it will be definitely make a fine contribution to the field and the wider community. In its present form, however, the manuscript appears, at least in parts, pre-liminary. This includes issues on clarity of writing, on presentation of results, and on the use of experimental models that are not stringently followed up to the end. In addition, I saw an earlier version of this manuscript submitted elsewhere previously and think that some of the earlier referees comments could have taken into account better. In addition, some of the results that the authors removed for this new submission would have been an asset to this present manuscript. I will specify below.

Major points

-The exchange of proof for nucleo-anterior proteolysis with the proposition of the presence of invadopodia in Figure 1a and d has not helped to convince the reader. Invadopodia -in case they are indeed present here - could in principle form all around the cell body. It is very much more difficult to proof their presence within a 3D collagen lattice as compared to a continuous flat gelatin coat (as shown by the group in Monteiro et al JCB 2013). To include invadopodia here takes the authors to 'shaky ground' which is not even necessary to enter as easier readouts are available. It would instead be much more advisable to show, in low and high density collagen, the presence and absence of both MT1-MMP-pHluorin (as shown in a previous manuscript version and in Monteiro et al) as well as collagen degradation (by COL1 $\frac{3}{4}$ Ab) signal anterior to the nucleus together with quantifications of multiple cells along the longitudinal cell body axis. Related to this, determining the number of invadopodia formation in Figures 4 d-g after modulation of Lis1, a linker protein of the nucleus-tubulin connection, seems not relevant to me. I have no doubts that a connection between Lis and invadopodia exists as measured by the authors, but it does not help the flow of the study: 1. there is no proof that invadopodia are indeed markers for proteolysis in the model used, 2. cells migrated on a thick but flat collagen surface and therefore did not 'feel' confinement, and 3. it is not clear from the images depicted where the front part and where the rear part of the cells is, thus it is not clear in which part of the cell the invadopodia are located. It would have helped to use again collagenolysis/ speed as a readout in both confining versus loose collagen, as this is the key readout for this study.

-Some figures should be harmonized for a better, logical order. As example, Figure 1e,f shows important initial findings, which are the reduction of collagen degradation at equal cell speeds in loose compared to dense collagen, together with changes in the presence of MMP inhibitor. Implementation of data after LMNA modulation already at this stage is confusing. Together with the text, it is hard to follow the authors here. Some reorganization of figures would greatly help.

- Actin contractility plays an important role in translocating the cell nucleus through a small pore (Thomas DG, et al. Non-muscle myosin IIB is critical for nuclear translocation during 3D invasion. J Cell Biol. 2015). Thus, what is the impact of the actin cytoskeleton in assisting to sort MT1-MMP positive endosomes anterior to the nucleus? Do tubulin and actin play here redundant roles? Even if the authors do not perform respective experiments they should at least discuss this issue.

-In general, the text, ie. the result part, appears very technical and makes it sometimes hard to follow the bigger lines of this study. i.e. is the part on Lis1 not always clear. Why is it important to show the proximity of Lis with lamin A/C, and would it be perhaps sufficient to show the

centrosome-nucleus distance after Lis modulation?

Other points

- some information is redundant, ie. invasion speed in the presence of MMP inhibitor GM6001 is shown in Figs 1e and S1d. Are these data derived from different invasion assays? The results are described in lines 99 and 120, but actually belong together. In another example, authors refer to the same Figure (S4a) for an increased centrosome-nucleus distance at two locations in the text (lines 208 and 231).
- the use of sps and lps for small and large pore sized collagen is colloquial and a more specific term should be used, but this might be an editorial issue.
- the LMNA gene encodes for the lamin A and lamin C protein, with lamin C being a splice variant. Therefore, the use of the term LMNA/C is wrong.
- the choice of colors in the different images is not always optimal. For example, a mix of pink with red, i.e. in Figure 1, does not allow a clear discrimination of structures, and the use of dark blue for nuclei or collagen precludes any optimal visualization of these structures.
- the display of results from less than 20 cells, as shown in Fig. 1e, should not be in box plots, but as single dots. As an alternative, the authors may simply add more cells to their analysis.
- in Fig. 3g, adding the cartoon on 'back' events (centrosomes not in front) is confusing, as these events were not counted.
- in some figures the axis legends are missing. Examples are Fig. 1c, S1e and I, etc.
- in some figures, controls are missing, such as in Fig. S1d,e, S2c/d.
- based on the data shown in Fig. S2c, it is possible to calculate a speed for GFP-LMNA cells.
- some sentences sound somewhat odd or are inaccurate. Examples are: line 144, 'Cells were cultured in a larger pore size collagen environment...'; line 194, '...is a component of the adaptive response to confining forces exerted on the nucleus' and it is not clear what confining forces are. Line 157 states '...consistent with reduced nuclear stiffness' which the authors did not measure, so at least a citation should be added.
- the authors mentioned that, but did not explain why, nocodazole treatment enhances the perinuclear Lis-1 distribution and what it actually means. Also, it is not entirely clear to me what the Lis-1/ lamin A/C proximity ligation assay is good for?
- what is the rationale for using spheroids in Figure 4b, when all other data are derived from single cells? Can the reduced collagen degradation in Fig. S4a be a result of impaired migration?

Reviewer #2 (Remarks to the Author):

I previously reviewed a version of this paper for NCB almost 2 years ago. The present version/revision adds data focussed on determining the role of Lis1 in collagenolysis and seems to have removed some experiments using p50/Dynamitin.

As I said before, I find the study interesting and largely convincing – and I still do.

I have a list of minor points which the authors could consider prior to publication:

1. The 3D projection in Movie 1 does not encompass the entire Z-axis of the cell. Thus, at present, it is not possible to determine whether a 'cage-like structure' is really formed.
2. What exactly constitutes a 'deformed nucleus' in regard of the scoring system used. The parameters described in Fig S1f (the authors refer to Fig. S1b in the methods in this regard, but I think that this is a mistake) seem rather arbitrary. Some sort of quantitative morphological assessment and/or assurance that scoring was performed blind would be helpful here.
3. There is no untreated control for Fig. S1g.
4. Could the authors display the collagen polymerisation temperatures used for the images in Fig. 1a?
5. The authors argue that Lis1 dependent coupling of the nucleus to the MT-network explains why Lis1 knockdown reduces invadopodia formation and collagen degradation. However, Lis1 has well described roles in vesicular trafficking so the observed decreases in invadopodia and degradation could be due to a role for Lis1 in vesicular transport along microtubules. To address this, the authors could assess the effects of Lis1 knockdown on MT1-MMP exocytosis in a 2D environment, (e.g. through imaging pHluorin-MT1-MMP in TIRF microscopy).
6. In figure 1f and 1 h the authors show that cells in lps-collagen have much lower levels degradation than cells in sps-collagen and argue that modulation of confinement by matrix correlates with changes in collagen degradation levels. However, these cells appear to be in contact with much less collagen (compare Figures 2a and 2d). The authors have not normalised their data to the collagen level per unit area which makes interpretation of collagen degradation levels difficult.
7. The images of phalloidin-stained spheroids in Fig. 4b are not very convincing
8. Typos on lines 239 and 307

Reviewer #3 (Remarks to the Author):

In this manuscript, the authors make a number of correlative findings. However, many of them are over interpreted. They attempt to describe mechanism, but fall short. The data are clear and straight forward. They show that changes in the lamin make up of the nuclear envelope changes the ability of the nucleus to move through small openings. These data are nice, but not novel, given other recent reports of similar findings. They then show that LINC complexes and Lis1 are also involved in the pathway. However, their roles could be quite indirect. Disruption of LINC or Lis1 are likely to lead to global cytoskeletal defects, and the exact mechanism is difficult to determine. Thus, the novelty of the findings here is not high, and the contributions to mechanism are not significant. The conclusions should be toned down, and the paper submitted to a more specialized journal.

Specific comments:

1. Figure 1 is not novel, as multiple similar reports have been published in the past couple of years.
2. They conclude in Fig 2 G-H that overexpression of laminA makes nuclei more stiff and act like they are in smaller collagen matrices. However, they do not control for the GFP tag on overexpressed laminA. Thus, is the phenotype because of overexpression, or the presence of GFP. Many in the field use GFP:lamin, but no one to my knowledge has shown this functions normally.
3. The statistics in Table 2 could easily be incorporated directly into the figures.
4. The dnKASH experiments are a blunt tool. It is likely that all nesprins are blocked from the nuclear periphery (not just nesprin1, as shown) and this can lead to all sorts of global cytoskeleton problems. The centrosome to nuclear distance defect described here is most likely due to a global cytoskeleton defect, not a specific LINC complex transferring forces across the nuclear envelope. The statement on lines 212-213 is way too strong. How many nesprins are expressed in these cells?
5. Line 236, centrosome at the back of the nucleus is misleading. Its not at the back, just behind

the leading edge.

6. Line 240, the role of Lis1 is over concluded. It could be a global cytoskeleton problem. It is not shown that the population of Lis1 at the nuclear envelope is the source of the polarization and centrosome positioning.

7. Line 263, Lis1 at the nuclear envelope could be very indirect of LINC.

RE : NCOMMS-17-06498

Reviewer #1 (Remarks to the Author):□

In the manuscript Infante et al the authors describe how proteolysis in substrate confinement is organized anterior to the bulky nucleus that -like a piston (now an established term by previous publications by Petrie & Yamada)- localizes MT1-MMP positive endosomes to its front, in order to have constricting substrate fibers degraded. This 'piston' mechanism only works in confinement and/ or if the nucleus is mechanically stiffened which the authors deduct from high versus low lamin A/C expression levels, as well as by an intact LINC-dynein-tubulin connection. The concept as such is very interesting, original and plausible and derives smoothly from earlier concepts in the field. If well presented, it will be definitely make a fine contribution to the field and the wider community. In its present form, however, the manuscript appears, at least in parts, preliminary. This includes issues on clarity of writing, on presentation of results, and on the use of experimental models□that are not stringently followed up to the end. In addition, I saw an earlier version of this manuscript submitted elsewhere previously and think that some of the earlier referees comments could have taken into account better. In addition, some of the results that the authors removed for this new submission would have been an asset to this present manuscript. I will specify below.□

R. We appreciate the reviewer positive comments and detailed and very valuable suggestions. Thanks to his/her critiques some experiments were done better and the manuscript has been reorganized and edited to improve clarity.

Major points

-The exchange of proof for nucleo-anterior proteolysis with the proposition of the presence of invadopodia in Figure 1a and d has not helped to convince the reader. Invadopodia -in case they are indeed present here - could in principle form all around the cell body. It is very much more difficult to proof their presence within a 3D collagen lattice as compared to a continuous flat gelatin coat (as shown by the group in Monteiro et al JCB 2013). To include invadopodia here takes the authors to 'shaky ground' which is not even necessary to enter as easier readouts are available. It would instead be much more advisable to show, in low and high density collagen, the presence and absence of both MT1-MMP-pHLuorin (as shown in a previous manuscript version and in Monteiro et al) as well as collagen degradation (by COL1 $\frac{3}{4}$ Ab) signal anterior to the nucleus together with quantifications of multiple cells along the longitudinal cell body axis.

R. We agree that MT1-MMP-pHLuorin could help documenting that focal degradation occurs at the nucleo-anterior region of the cell. Although the MT1-MMP-pHLuorin construct that we developed and used in our previous work is

ideal to show exocytic events of MT1-MMP-positive storage endosomes, especially on a flat fibrous collagen (or gelatin) layer, it is more problematic to use in a 3D environment. Probably due to overexpression conditions, MT1-MMP-pHLuorin tends to lose its polarized surface-distribution over time as it accumulates at the cell surface. Instead, to overcome issues due to overexpression, we have monitored collagen degradation by Col1-^{3/4}C Ab staining in cells expressing endogenous MT1-MMP level in low pore size (Fig. 1d-h) vs. large pore size collagen (Fig. 1i). Cells were incubated for a short period of time (2.5 hrs) in the collagen gels to avoid cumulative degradation signal over time and then fixed and stained. Very consistently, we observed three subpopulations of cells in the small pore size gel based on their morphology: (1) round cells that did not initiate migration - yet - with pericellular matrix degradation all over (as in Fig. 1d), (2) cells that initiated polarization and invasion with a short protrusion and with matrix degradation visible at the basis of the protrusion ahead of the nucleus (Fig. 1e), (3) fully polarized invasive cells with a long protrusion and collagen degradation visible mainly in association with the bulky cell region anterior to the nucleus (Fig. 1f). On the contrary, pericellular collagen degradation was limited when cells were incubated in the large pore size gel (Fig. 1i). In order to visualize the presence of invadopodia in the 3D collagen network, we co-stained cells with phalloidin to label F-actin, the main invadopodia component. Panels (g) and (h) of the revised Figure 1 show collagen degradation (Col1-^{3/4}C) in association with F-actin-rich invadopodia forming in contact with the fibrils ahead of the nucleus.

Related to this, determining the number of invadopodia formation in Figures 4 d-g after modulation of Lis1, a linker protein of the nucleus-tubulin connection, seems not relevant to me. I have no doubts that a connection between Lis and invadopodia exists as measured by the authors, but it does not help the flow of the study

R. We respectfully disagree and we think that the contribution of the nucleus-microtubule linkage and the role of LINC complex and Lis1 are equally important for the mechanism of invadopodia-based matrix degradation during the confined migration of tumor cells in the 3D collagen environment. This statement is based on the fact that knockdown of Lis1 has several effects, which are similar to the inhibition of LINC complex function by overexpression of the dominant inhibitory KASH construct or by knockdown of nesprin-2.

- nucleus-centrosome linkage is affected (nucleus-centrosome distance is increased from 0.33 μm in control siNT-treated cells to 1.44-2.03 μm in siLis1-depleted cells, see Fig. 3e)
- centrosome positioning is affected (see Fig. 3e)
- pericellular collagenolysis by MDA-MB-231 and HT-1080 cells in 3D collagen is reduced by ~60% as compared to control siNT-treated cells (Fig. 4b)
- it interferes with MT1-MMP-endosome polarization in the nucleo-anterior region (Fig. 4c)

- it reduces the formation of invadopodia measured by the recruitment of the invadopodial protein Tks5 at plasma membrane-collagen fibril contact sites both in MDA-MB-231 (reduced to 60-70% of control level) and HT-1080 cells (reduced to ~50% of control levels) (Fig. 4e).

In addition, we found that a fraction of the Lis1 protein associates with the nuclear envelope and that this association is inhibited by overexpression of the dominant inhibitory KASH construct and by knockdown of nesprin-2, arguing for a functional link between nesprin-2 and Lis1 proteins and their contribution to a common mechanism.

1. there is no proof that invadopodia are indeed markers for proteolysis in the model used

R. We now provide better images showing that collagen degradation (visualized with Col1-^{3/4}C Ab) is associated with F-actin-rich structures forming in contact with confining collagen fibrils in the nucleus-anterior region during confined migration in the 3D collagen environment (see Fig. 1gh). These structures match with the operational definition of invadopodia at the light microscopy level proposed by Bowden et al. (Methods Cell Biol. 63:613-, 2001), which is the localization of a series of proteins (e.g., F-actin, cortactin, Tks5...) at sites of degradation.

2. cells migrated on a thick but flat collagen surface and therefore did not 'feel' confinement

R. We agree that it may seem paradoxical that interfering with LINC complex or Lis1 function affects invadopodia formation in cells cultured on top of a 2D collagen layer. Here, we would like to make the parallel with macrophages that attempt to engulf the glass surface to which they adhere to in a process called frustrated phagocytosis. We think that tumor cells probably interpret the flat collagen layer as a bona fide matrix, and this switches the MT1-MMP/invadopodia response on at the ventral cell surface in a LINC complex/Lis1-dependent manner. This issue is discussed in the discussion section of the revised version of the manuscript (lines 331-336).

3. it is not clear from the images depicted where the front part and where the rear part of the cells is, thus it is not clear in which part of the cell the invadopodia are located. It would have helped to use again collagenolysis/ speed as a readout in both confining versus loose collagen, as this is the key readout for this study.

R. In the 3D collagen network it is reasonably easy to infer the direction of migration of the cells before fixation based on their shape, the presence of nucleus at the cell rear and the cell protrusion at the front. In addition, in the images provided in Fig. 1, cells in panel d-f and in panel (i) have been stained for alpha-tubulin to visualize the centrosome (microtubule-organizing center), which is ahead of the nucleus in the direction of migration (of note, available pericentrin mAb did not work in the 3D collagen gel for co-labeling with Col1-^{3/4}C polyclonal Ab). In addition, Col1-^{3/4}C staining that reveals cumulative collagen degradation during the 2.5 hrs incubation time is also showing the direction of cell movement from the initial position of the cell visualized by the cleaved fibrils to its position at the time of fixation. When cells are plated on top of a thick collagen layer and fixed as shown in Fig. 4, the direction of migration is unknown. However, here the nucleus-to-cell ventral surface axis is probably more relevant for invadopodia formation.

- Some figures should be harmonized for a better, logical order. As example, Figure 1e,f shows important initial findings, which are the reduction of collagen degradation at equal cell speeds in loose compared to dense collagen, together with changes in the presence of MMP inhibitor. Implementation of data after LMNA modulation already at this stage is confusing. Together with the text, it is hard to follow the authors here. Some reorganization of figures would greatly help.

R. We thank the referee for his/her comment and suggestion. LMNA modulation data have been moved from Figure 1 to Figure 2. Figure 3 was also reorganized (see below).

Actin contractility plays an important role in translocating the cell nucleus through a small pore (Thomas DG, et al. Non-muscle myosin IIB is critical for nuclear translocation during 3D invasion. J Cell Biol. 2015).

Thus, what is the impact of the actin cytoskeleton in assisting to sort MT1-MMP positive endosomes anterior to the nucleus? Do tubulin and actin play here redundant roles? Even if the authors do not perform respective experiments they should at least discuss this issue.

R. We do not want to minimize the critical role played by actomyosin during cell migration in confined 3D environments. The important work by Thomas *et al* as well as other studies is discussed in the discussion section of our manuscript. However, our study focused on some novel aspects of the mechanism of confined migration depending on the microtubule system and nucleus-centrosome linkage. We agree with this referee that it will be necessary to put the contribution of tubulin and actin together, but we believe that it is out of the scope of the present study.

8. In general, the text, ie. the result part, appears very technical and makes it sometimes hard to follow the bigger lines of this study. i.e. is the part on Lis1 not always clear. Why is it important to show the proximity of Lis with lamin A/C, and would it be perhaps sufficient to show the centrosome-nucleus distance after Lis modulation?

R. We thank the referee for his/her comments and suggestions to improve the clarity of the manuscript. The Result section of the revised manuscript was modified to make it less technical. Regarding the part on Lis1, data have been reorganized (Figure 3). Laser ablation experiments documenting the existence of some tension force in the nucleus-centrosome axis during confined migration of MDA-MB-231 cells is shown first (Fig. 3ef), then Lis1 association with the nuclear envelope is reported (Fig. 3gh) and finally the effects of Lis1 knockdown on nucleus-centrosome distance and centrosome localization are provided (Fig. 3i and j). Data showing the proximity of Lis1 with LMNA have been deleted.

Other points

i. some information is redundant, ie. invasion speed in the presence of MMP inhibitor GM6001 is shown in Figs 1e and S1d. Are these data derived from different invasion assays? The results are described in lines 99 and 120, but actually belong together.

R. We apologize for data duplication. Duplicated data concerning invasion in the presence of GM6001 have been deleted (see Supplementary Fig. 1).

ii. In another example, authors refer to the same Figure (S4a) for an increased centrosome-nucleus distance at two locations in the text (lines 208 and 231).

R. Duplication has been deleted.

iii. the use of sps and lps for small and large pore sized collagen is colloquial and a more specific term should be used, but this might be an editorial issue.

R. In the revised text, we refer to “the small pore size collagen gel” to the gel obtained by polymerization at 37°C, and “high pore size gel” for the one polymerized at 20°C and clear indications of polymerization temperatures are provided in the figure legends.

iv. the LMNA gene encodes for the lamin A and lamin C protein, with lamin C being a splice variant. Therefore, the use of the term LMNA/C is wrong.

R. We now refer to LMNA.

v. the choice of colors in the different images is not always optimal. For example, a mix of pink with red, i.e. in Figure 1, does not allow a clear discrimination of structures, and the use of dark blue for nuclei or collagen precludes any optimal visualization of these structures.

R. Whenever possible, colors have been changed and we have used color combination to facilitate visualization of the different proteins and dyes. For several figures, as Fig. 1a, Fig. 1e/f, Fig. 3g, Fig. 4d ... insets showing separated channels are also provided.

vi- the display of results from less than 20 cells, as shown in Fig. 1e, should not be in box plots, but as single dots. As an alternative, the authors may simply add more cells to their analysis.

R. Data are plotted as individual dots whenever $n < 20$ in the revised manuscript.

vii- in Fig. 3g, adding the cartoon on 'back' events (centrosomes not in front) is confusing, as these events were not counted.

R. Cartoon on "back" events in Fig. 3j (revised manuscript) has been deleted.

viii- in some figures the axis legends are missing. Examples are Fig. 1c, S1e and I, etc.

R. Axis legends have been added when missing (Fig. 1c, Fig. S3d, Fig. S4f)

ix- in some figures, controls are missing, such as in Fig. S1d,e, S2c/d.

R. Due to duplicated data (see point (i) raised by this referee), data in Fig. S1d/e have been deleted in the revised version of the manuscript.

In Fig. S2c/d, we are comparing the transmigration of nuclei of ^{GFP}H2B-expressing cells treated with siNT or siLMNA. This information is now clearly displayed in the legend of Fig. 2c/d. We agree that there is no former control to compare to the transmigration of cells expressing ^{GFP}LMNA (except the ^{GFP}H2B/siNT cells, which are not ideal). However, overexpression of ^{GFP}LMNA in MDA-MB-231 cells resulted

in the expected phenotype, i.e. nucleus were stuck in the narrow 2.5 μ m-diameter constriction).

x- based on the data shown in Fig. S2c, it is possible to calculate a speed for GFP-LMNA cells.

R. Out of thirteen ^{GFP}LMNA-positive nuclei analyzed, two nuclei passed through and eleven nuclei stalled within the 2.5 μ m-diameter constriction during the 12-hr movie; thus speed could not be calculated. This information is mentioned in the legend of Fig. S2d.

xi- some sentences sound somewhat odd or are unaccurate. Examples are: line 144, 'Cells were cultured in a larger pore size collagen environment...'; line 194, '...is a component of the adaptive response to confining forces exerted on the nucleus' and it is not clear what confining forces are. Line 157 states '...consistent with reduced nuclear stiffness' which the authors did not measure, so at least a citation should be added.

R. The manuscript has been edited.

'Cells were cultured in a larger pore size collagen environment...'
replaced by:

'Matrix pore size was modulated by inducing collagen polymerization at 20°C while keeping the collagen concentration constant (2.0 mg/ml), leading to a ~2-fold increase in the distance between fibrils as compared to polymerization performed at 37°C, the condition used so far (Supplementary Fig. 1ef)...' (lines 115-118)

'...is a component of the adaptive response to confining forces exerted on the nucleus'

replaced by:

'Collectively these findings suggested that endosome polarization in front of the nucleus is part of the response to adjust pericellular collagenolysis to changing ECM environments and widen constricting matrix pores.' (lines 193-196)

'...consistent with reduced nuclear stiffness'

This sentence has been deleted in the revised version.

xii- the authors mentioned that, but did not explain why, nocodazole treatment enhances the perinuclear Lis-1 distribution and what it actually means.

R. It has been previously reported that Nocodazole treatment induces Lis1 enrichment at the nuclear envelope (NE), allowing better visualization in different cell types (Smith et al. Nat Cell Biol 2:767-, 2000; Coquelle et al. Mol Cell Biol 22:3089-, 2002; Baffet et al. Dev Cell 33:703-, 2015). To our knowledge, the mechanism underlying enhanced Lis1 enrichment at the NE by nocodazole is unknown. The conclusion is that Lis1 and dynein function at the NE to mediate nucleus-centrosome linkage.

Also, it is not entirely clear to me what the Lis-1/ lamin A/C proximity ligation assay is good for?

R. Lis1/LMNA PLA data have been deleted in the revised manuscript. Fig. 3g/h shows the distribution of Lis1 in MDA-MB-231 cells and its association with the nuclear membrane and the contribution of LINC complex and nesprin-2.

xiii- what is the rationale for using spheroids in Figure 4b, when all other data are derived from single cells?

R. In Fig. 4a, multicellular spheroid data in small pore size collagen gel have been deleted in the revised version and replaced by single cell analysis as in Fig. 1j. The two assays showed similar inhibitory effect of Lis1 knockdown on invasion. We also included new data in Fig. 4a documenting that Lis1 KD does not affect invasive migration in the large pore size collagen gel.

Can the reduced collagen degradation in Fig. S4a be a result of impaired migration?

R. It does not seem that cells need to migrate in order to degrade the surrounding collagen. For instance, see the round cell in Fig. 1d that did not initiate invasive migration yet but show high level of matrix degradation all around. On the contrary, polarized matrix degradation is necessary for confined migration in the small pore size collagen gel.

Reviewer #2 (Remarks to the Author):

I previously reviewed a version of this paper for NCB almost 2 years ago. The present version/revision adds data focussed on determining the role of Lis1 in collagenolysis and seems to have removed some experiments using p50/Dynamitin.

As I said before, I find the study interesting and largely convincing – and I still do.

R. We thank the referee for his/her detailed analysis of our manuscript and positive comments. Because of space limitation and because they were outside the main focus of the study, we have decided to remove the p50/Dynamitin data. We would like also to mention that we recently published a article addressing the function of microtubule-based motors including the dynein/dynactin complex in MT1-MMP endosome trafficking and its impact on the invadopodia/MT1-MMP response (Marchesin et al. J Cell Biol 211:339-, 2015).

Minor points:

1. The 3D projection in Movie 1 does not encompass the entire Z-axis of the cell. Thus, at present, it is not possible to determine whether a ‘cage-like structure’ is really formed.

R. We are now providing a movie showing the entire Z-stack of the cell (Movie S1).

2. What exactly constitutes a ‘deformed nucleus’ in regard of the scoring system used. The parameters described in Fig S1f (the authors refer to Fig. S1b in the methods in this regard, but I think that this is a mistake) seem rather arbitrary. Some sort of quantitative morphological assessment and/or assurance that scoring was performed blind would be helpful here.

R. The reference figure for ‘deformed nucleus’ scoring system is shown in Supplementary Fig. S1a in the revised manuscript. Nuclei were classified using the same descriptors for nucleus shape (‘hourglass’, ‘prolapse’ ...) as in Wolf et al (J Cell Biol 201: 1069-, 2013) but for simplification and as not being the major point of this work not round nuclei were all merged to a single group named deformed. Subgroups quantification (hourglass, prolapse....) can be provided upon request.

3. There is no untreated control for Fig. S1g.

R. Untreated control has been added in Supplementary Fig. S1d of the revised manuscript (Fig. S1g in the first submission).

4. Could the authors display the collagen polymerisation temperatures used for the images in Fig. 1a?

R. Temperatures of collagen polymerization are displayed in the legend of Fig.1 in this revised version.

5. The authors argue that Lis1 dependent coupling of the nucleus to the MT-network explains why Lis1 knockdown reduces invadopodia formation and collagen degradation. However, Lis1 has well described roles in vesicular trafficking so the observed decreases in invadopodia and degradation could be due to a role for Lis1 in vesicular transport along microtubules. To address this, the authors could assess the effects of Lis1 knockdown on MT1-MMP exocytosis in a 2D environment, (e.g. through imaging pHluorin-MT1-MMP in TIRF microscopy).

R. This comment is well taken, although Lis1 involvement in vesicular traffic has been controversial. It was shown that expression of dominant negative Lis1 construct inhibits mitosis and cell migration, with no effect on lysosome, endosome or Golgi distribution (Faulkner *et al.*, Nat Cell Biol 11:784-, 2000; Dujardin *et al.* J Cell Biol 163:1205-, 2003). However, in other studies, Lis1 overexpression caused Golgi compaction (Smith *et al.* Nat Cell Biol 2:767, 2000) and Lis1 knockdown was reported to disperse some vesicular organelles (Lam *et al.* J Cell Sci 123:202, 2010). Thus, implications of these disparate observations for Lis1 function in vesicular transport remain unclear. Some more recent studies concluded that a role of Lis1 in organelle transport may depend on the size of the load (organelle) as well as the cell type being studied (Pandey and Smith J Neurosci 31:17207-, 2011; Yi *et al.* J Cell Biol 195:193-, 2011; Reddy *et al.* Nat Comm 7:12259-, 2016).

We previously showed that trafficking and positioning of MT1-MMP-positive endosomes depend on the dynein/dynactin complex in MDA-MB-231 cells (Marchesin *et al.* J Cell Biol 211:339-, 2015). Thus, we expressed the p50/Dynamitin subunit known to disrupt the dynactin complex resulting in anterograde movement of endosomes through unbalanced kinesin activity (Burkhardt *et al.* J Cell Biol 139:469-, 1997). The effect of p50/Dynamitin on MT1-MMP endosome positioning was compared to that of Lis1 knockdown. The results of this analysis (provided in Supplementary Fig. 5d of the revised manuscript), show the expected dramatic peripheral redistribution of MT1-MMP-positive endosomes upon p50/Dynamitin expression while the overall distribution of MT1-MMP endosomes is not perturbed upon Lis1 silencing. Thus we conclude that Lis1 does not seem to regulate dynein activity in MT1-MMP endosome trafficking, while the dynactin complex does.

6. In figure 1f and 1 h the authors show that cells in lps-collagen have much lower levels degradation than cells in sps-collagen and argue that modulation of confinement by matrix correlates with changes in collagen degradation levels. However, these cells appear to be in contact with much less collagen (compare Figures 2a and 2d). The authors have not normalised their data to the collagen level per unit area which makes interpretation of collagen degradation levels difficult.

R. The concentration of collagen was kept constant throughout the study (2.0 mg/ml) and two temperatures of polymerization were used to modulate pore size (37°C and 20°C). The conclusion that MT1-MMP-dependent collagenolysis is an adaptive response is also supported by the fact that overexpression of ^{GFP}LMNA in MDA-MB-231 cells increases collagenolysis in the large pore size collagen gel (polymerized at 20°C) to levels similar those observed with the same cells in the small pore size gel (polymerized at 37°C) (Fig. 2b); and by the observation that silencing of LMNA decreased significantly collagenolysis levels in the small pore size gel as compared to the non-silenced cells in the same collagen environment (Fig. 2c).

7. The images of phalloidin-stained spheroids in Fig. 4b are not very convincing ?

R. To homogenize the conditions used to assay the invasive capacity of the different cell populations, data based on the multicellular spheroid assay (Fig. 4b/c of the initial submission) have been replaced by a single cell-based assay as in Fig. 1j. The two assays showed similar inhibitory effect of Lis1 knockdown on invasion.

8. Typos on lines 239 and 307

*239 of centrosome mispositioning (Fig. 3h and Supplementary Movie 7). All together
307 type I collagen (here). All together, our findings indicate that interfering with nuclear*

R. Typos have been corrected.

Reviewer #3 (Remarks to the Author):

In this manuscript, the authors make a number of correlative findings. However, many of them are over interpreted. They attempt to describe mechanism, but fall short. The data are clear and straight forward. They show that changes in the lamin make up of the nuclear envelope changes the ability of the nucleus to move through small openings. These data are nice, but not novel, given other recent reports of similar findings. They then show that LINC complexes and Lis1 are also involved in the pathway. However, their roles could be quite indirect. Disruption of LINC or Lis1 are likely to lead to global cytoskeletal defects, and the exact mechanism is difficult to determine. Thus, the novelty of the findings here is not high, and the contributions to mechanism are not significant. The conclusions should be toned down, and the paper submitted to a more specialized journal.

R. We thank the referee for his/her critical assessment of our manuscript. We respectfully disagree regarding his/her comments that question the novelty of our findings. We believe that the concept of nucleus-induced stress for ECM proteolysis and confined migration is novel and highly relevant for the field explaining fundamental concepts of cell migration, especially in light of the new attention towards the effect of confined substrate geometries or nuclear deformability on migration.

Specific comments:

1. Figure 1 is not novel, as multiple similar reports have been published in the past couple of years.

R. The mechanism we propose, which derives from and integrate earlier concepts in the field is highly novel and original. Some of the data displayed in Fig. 1 corroborate previous findings obtained in different cell types and in various experimental conditions. Our study gives a new perspective as it integrates a quantitative assessment of collagenolysis and reveals modulations based on collagen gel porosity and LMNA levels. Fig. 1 also provides new information regarding the part of the cell in which invadopodia are located and where collagenolysis mostly occurs, i.e. at the nucleo-anterior bulky part of cell.

2. They conclude in Fig 2 G-H that overexpression of laminA makes nuclei more stiff and act like they are in smaller collagen matrices. However, they do not control for the GFP tag on overexpressed laminA. Thus, is the phenotype because of overexpression, or the presence of GFP. Many in the field use GFP:lamin, but no one to my knowledge has shown this functions normally.

R. GFP-LMNA is indeed widely used in the field and has become the standard. As

described by others, we see that GFP-tagged LMNA dramatically slows down and even blocks the transmigration of the nucleus and the cell in the 2.5 um-diameter constriction of the micro-fabricated device we used in contrast to cells overexpressing nuclear GFP-tagged H2B used as a control. Therefore, the GFP-tag as such does not seem to perturb the migration of the cells in the narrow constriction when fused to H2B.

3. The statistics in Table 2 could easily be incorporated directly into the figures.

R. As suggested by this referee, statistics in Table 2 have been incorporated in the corresponding Figures.

4. The dnKASH experiments are a blunt tool. It is likely that all nesprins are blocked from the nuclear periphery (not just nesprin1, as shown) and this can lead to all sorts of global cytoskeleton problems. The centrosome to nuclear distance defect described here is most likely due to a global cytoskeleton defect, not a specific LINC complex transferring forces across the nuclear envelope.

R. We have looked at the effect of DN-KASH on Nesprin-2 and Nesprin-1 distribution in MDA-MB-231 cells and both were affected (in contrast to SUN1, see Supplementary Fig. S4b/c/d of the revised manuscript). We did not look at the other nesprins. Using several assays, we obtained similar effects of DN-KASH expression and nesprin-2 silencing such as on Lis1 association with the nuclear envelope (Fig. 3h), centrosome to nuclear distance defect (Fig. 3i and Supplementary Fig. S4a, with stronger DN-KASH effects) and invadopodia formation in association with the collagen fibrils (Fig. 4e/f). In addition, we have looked at the capacity of cells expressing DN-KASH to assemble cortical actin-based cytoskeletal structures. As shown in Supplementary Fig. S6, DN-KASH expressing cells retain full capacity to assemble actin-based lamellipodia and to spread on a 2D substratum in response to constitutively active Rac1L61 mutant.

The statement on lines 212-213 is way too strong. How many nesprins are expressed in these cells?

R. The sentence has been modified:

'All together, these findings suggested that perturbation of LINC complex function and of nucleus-cytoskeletal linkage interfered with front polarization of MT1-MMP storage compartments and with pericellular collagenolysis during confined migration in 3D.' (lines 217-219)

5. Line 236, centrosome at the back of the nucleus is misleading. Its not at the back, just

behind the leading edge.

R. Based on the analysis of movies of MDA-MB-231 cells expressing GFP-Centrin-1 invading through the small pore size collagen gel, we found a predominant localization of the centrosome in front of the nucleus. In contrast, the centrosome was positioned at behind the nucleus front edge in cells knocked down for Lis1 (see Fig. 3j). We toned down our conclusion and stated that ‘interfering with Lis1 function affected centrosome positioning in front of the nucleus during confined migration in the collagen matrix, suggesting a role for Lis1 in nucleus-centrosome linkage in MDA-MB-231 cells.’ (lines 254-256)

6. Line 240, the role of Lis1 is over concluded. It could be a global cytoskeleton problem. It is not shown that the population of Lis1 at the nuclear envelope is the source of the polarization and centrosome positioning.

R. Based on the observation that Lis1 is associated with the nuclear envelope (Fig. 3g/h) and Lis1 silencing induces a centrosome to nuclear distance defect (Fig. 3i) and mispositioning of the centrosome at the back of the cell during invasive migration (Fig. 3j), we concluded that Lis1 may control nucleus-centrosome linkage and centrosome positioning. Analysis of neuronal cells defective for Lis1 supported similar conclusions (Tanaka et al. J Cell Biol 165:709-, 2004; Shu et al Neuron 44:263-, 2004).

Yet, when they were embedded in the 3D large pore size collagen gel, Lis1 knocked down cells invaded with a speed similar to that of cells treated with an irrelevant siRNA (Fig. 4a).

Moreover, in the revised manuscript, we provide new data in response to Point #5 raised by referee 2. These experiments showed that the overall distribution of MT1-MMP-positive endosomes, which depends on their interaction with the microtubule network (Marchesin et al. J Cell Biol 211:339-, 2015), was not affected by Lis1 silencing (see Supplementary Fig. S5d). All together, these observations argue against a global cytoskeleton problem by interfering with Lis1.

7. Line 263, Lis1 at the nuclear envelope could be very indirect of LINC.

R. We agree and actually did not state that Lis1 association was directly under LINC control. We observed that a fraction of the Lis1 protein associated with the nuclear envelope and that this association was inhibited by overexpression of the dominant inhibitory KASH construct or by knockdown of nesprin-2 (Fig. 3g/h). These findings are suggestive of a functional link between nesprin-2 and Lis1 proteins and a possible function for Lis1 at the nuclear envelope. (lines 240-241)

Reviewers' comments:

Reviewer #1 (Remarks to the Author):

The manuscript Infante et al. aims to dissect a physical / molecular mechanism on localized matrix degradation during migration in confinement. The concept described in the manuscript is exciting and will close a gap in the existing literature on an additional function of the nucleus during migration that, like a 'piston', directs MT1-MMP positive endosomes in its front. When these endosomes reach the cell surface, they get in contact with confining substrate fibrils that they degrade which will ease cell migration. The manuscript has improved since its last submission, however, is not smooth yet. There are still some sub-concepts that I find problematic, and the manuscript partially comes across somewhat preliminary (sometimes unaccurate in text and figures and occasionally hard to read), examples are provided below. However, I believe with the correct adjustments this will become a fine paper that benefits the scientific readership in the Nature Communications journal.

Major points

1. It is not understandable to me why the authors focus so much on invadopodia. Even though it has been shown that these structures carry MT1-MMP and that they perform dot-like proteolysis on 2D substrate, their proteolytic involvement in 3D collagen degradation was not proven yet and most probably plays a minor effect in the overall ECM contact-dependent proteolysis. Also, invadopodia form everywhere around the cell and not just at locations of constrictions. In addition, the authors do not show whether MT1-MMP only localizes at the nucleo-anterior invadopodia during constricted migration. Gladly, the authors now focus on the product of overall collagenolytic activity, as by performing COL1 $\frac{3}{4}$ staining in initially migrating cells. Collagen degradation occurs wherever fibrils impinge but –more importantly- constrain the cell body. This takes place at locations of diameter/volume increase, i.e. at pseudopodial bifurcations or in front of the cell nucleus. The authors (anecdotal?) own images in Fig. 1e-h support this simple concept which, however, still needs to be quantified in some way (i.e. measuring the actin/ COL1 $\frac{3}{4}$ signal along the cell axis of a number of cells, as I already asked in my last review). This way the authors could generate a useful readout. – In contrast, the entire Figure 1 does not show a single structure that I would call an 'invadopod'. The actin rich spots in Fig. 1a are located at 'cell-matrix interaction sides' clarified by the white arrowheads. To name these actin locations 'invadopodia' is highly speculative.

Related to this, as I said before, I believe a connection between Lis and invadopodia, but it does not help the study for the above named reasons. The invadopodia marker stainings in Fig. 4d do not prove at all any MT1-MMP location or collagen degradation during constricted migration. In addition, these images have been acquired ON and not IN collagen, which proves the authors statement in the abstract 'whereby *mechanical constraints* on the nucleus trigger ... invadopodia formation ... in front of the nucleus depending on nucleus-microtubule linkage' wrong. Together, as advice I suggest that the authors exchange their Fig.4d-e with the more correct readout of COL1 $\frac{3}{4}$ localization of cells IN collagen after the same indicated treatments.

2. It is not clear why the authors do not discuss the piston mechanism as it fits wonderfully into their concept. Even though they cite the relevant papers (Refs 64,65) for another reason, they do not reflect on these interesting similar phenomenons.

3. As said, I generally support the authors concept that the Lis-LINC connection drives nucleus microtubule organization and MT1-MMP localization reflected by reduced invasion rates after Lis knockdown (Fig. 4a). However, this reduction is around 30% a compared to a 50% reduction after

GM6001 treatment (Fig. 1a). The authors should therefore discuss possible compensatory effects, i.e. by other nucleo-cytoskeletal connections i.e. actin, or others. Also, as an important final control, is the Lis defect restored in collagen with wide pores?

4. The authors change lamin A levels by downmodulation or overexpression. It has been previously shown that lamin A changes modulate nuclear stiffness. The authors, however, do not show it in this manuscript. They should therefore be careful with their statements (as mentioned in my last review). Instead of stating in the abstract 'Here, we show that modulation of matrix pore size or nuclear stiffness directly impinges on levels of MT1-MMP-mediated pericellular collagenolysis by cancer cells' the statement needs to be like 'Here, we show that modulation of matrix pore size or *lamin A expression known to modulate nuclear stiffness* directly impinges on levels of MT1-MMP-mediated pericellular collagenolysis by cancer cells'. This has to be revised through the entire text.

Other points

1. GM6001 is a general MMP inhibitor, and the authors should refer to GM6001 as such (in lanes 104, 106).
2. The presentation of results sometimes appears fractionated and results could be summarized: (A) i.e. Figures 1k and S1b could be merged to one figure and shown next to Fig. 1j - the reader would understand better the interesting point that migration in 20C collagen is fast despite only little proteolysis which is an important point guiding the reader to the central dogma of this study. The measurement of the proteolysis level in the presence of GM6001 is missing. (B) Figs 2A and S2c,d belong together and it is a pity that results were acquired in different migration systems. Perhaps the authors could think of a way to summarize the results into one figure. In this regard, in Fig. S2d, the authors should find a way to quantify the migration speed of GFP-LMNA cells even if it is very slow/ nucleus eventually stalls. (C) Figs. 3i and S4a belong together and should be supported by corresponding image examples.
3. The time annotation in Figs. 2e,h,k and 3a does not fit to their corresponding movies.
4. In Fig. S1a, the nuclear morphology showing a 'prolapse' rather looks like a nucleus with a micronucleus. I recommend to take it out and exclude such images from analysis.
5. There are text passages that do not read entirely accurately, i.e.:

-Lane 83 'F-actin enrichments were visible ahead of the nucleus in association with ECM fibrils oriented orthogonally to the direction of migration, probably opposing cell movement' Reads as if F-actin opposes cell movement

-lane 110 'All together, these findings indicated that surface-exposed MT1- MMP enabled confined nuclear and cell movement' reads as if MT1-MMP drives nuclear movement which the authors did not really investigate and which i.e. also appears in the absence of MT1-MMP activity (Fig. 1c). A better sentence here could be: 'All together, these findings indicated that surface-exposed MT1-MMP enabled efficient cell movement in confinement by mediating proteolysis anterior of the nucleus.'

-lane 119 'In the large pore size collagen network, the cell body was generally larger and more massive and the nucleus was positioned close to the cell center (Fig. 1i)'. For such a statement the authors need add some quantification.

-lane 311 The statement in the discussion 'Our data reveal an unprecedented mechanotransduction pathway in which tumor cells adapt to the 3D matrix environment by sensing constraints imposed on the nucleus and...'. Can the phenomenon described here really be called a mechanotransduction pathway which typically involves signaling molecule activation by some means of mechanostimulation? The authors may tone down this statement to i.e. 'Our data reveal a *novel mechanical mechanism* in which tumor cells adapt to the 3D matrix environment by sensing constraints imposed on the nucleus and...'

-lane 322, reads confusing 'The present study suggests that nucleus pulling force generation and the multistep invadopodia assembly program may be two sides of the same coin, governed by nucleus-cortex mechanical coupling through the microtubule network'. Pulling forces should be generated by the actin cytoskeleton, or better specified what pulling forces mean in the context of the microtubulus network.

Reviewer #2 (Remarks to the Author):

The revision of this paper from Infante et al. is improved. As I said before, I feel that the paper is largely acceptable for publication without the addition of any further experimentation. However, the way that paper is presented makes it quite tough reading. In particular, I think that the novelty of the main story is somewhat obscured by the inclusion of data that confirm previous findings and this is maybe why one of the other reviewers is struggling with the paper. Furthermore, some of the figure formatting and presentation could be improved to render the paper easier to read and thus increase its impact.

Specific comments on presentation:

1. Fig. 1g: I feel that the inserts are inadequate as they stand as their resolution is insufficient. If the point of these is to show association of col13/4 with actin, it would be preferable to show an actual quantification – or a much higher resolution image – or both.
2. I feel that all of figure 1 would be better relegated to supplementary. This is because, most – if not all – of these observations have been published before in one way or another, and this is if perhaps why one of the other reviewers is struggling with seeing the novelty of this paper. A better first figure would be to present the different conditions showing the links between confinement, nuclear deformation and invasion. This would make a better exposition of the key rationale for the rest of the paper.
3. Fig 3g would be more convincing if there were some colocalisation demonstrated between Lis 1 and a nuclear envelope marker.
4. GM6001 is not an specific MT1-MMP inhibitor – especially at 40 μ M. This assertion needs to be corrected in the text and mention made of the experiments performed with siRNA of MT1-MMP.
5. Should the x-axis of Fig. S1f have a label – say, 'interfibrillar distance'
6. I don't understand why the data showing nesprin2 displacement after DN-KASH expression is a a format that differs from that used to display the nesprin1 data (sup. Fig. 4b &c). Also, what is Fig. S4d attempting to show? – it appears quite confusing to me.
7. Where is the GFP in Fig. S6?

Re : NCOMMS-17-06498A

Reviewer #1 (Remarks to the Author):

The manuscript Infante et al. aims to dissect a physical / molecular mechanism on localized matrix degradation during migration in confinement. The concept described in the manuscript is exciting and will close a gap in the existing literature on an additional function of the nucleus during migration that, like a 'piston', directs MT1-MMP positive endosomes in its front. When these endosomes reach the cell surface, they get in contact with confining substrate fibrils that they degrade which will ease cell migration. The manuscript has improved since its last submission, however, is not smooth yet. There are still some sub-concepts that I find problematic, and the manuscript partially comes across somewhat preliminary (sometimes unaccurate in text and figures and occasionally hard to read), examples are provided below. However, I believe with the correct adjustments this will become a fine paper that benefits the scientific readership in the Nature Communications journal.

R. We do appreciate the reviewer's positive comments and his/her efforts to help improving our manuscript.

Major points

It is not understandable to me why the authors focus so much on invadopodia. Even though it has been shown that these structures carry MT1-MMP and that they perform dot-like proteolysis on 2D substrate, their proteolytic involvement in 3D collagen degradation was not proven yet and most probably plays a minor effect in the overall ECM contact-dependent proteolysis. Also, invadopodia form everywhere around the cell and not just at locations of constrictions.

R. As suggested by the referee, we diminished the emphasis on invadopodia in the revised version of the manuscript; the relationship of the on-demand collagenolysis response with invadopodia is analyzed and discussed at the end of the manuscript (see Fig. 5defg). We believe that invadopodia are instrumental in the context of matrix proteolysis including in the fibrous 3D collagen environment. We acknowledge that data showing invadopodial matrix degradation activity towards fibrous collagen are more limited than that of the 2D gelatin substrate. Yet, some published data exist indicating that invadopodia forming in association with collagen fibrils are degradative (Juin et al. MBoC 23 (2012) 297-309, Ezzoukhry et al. EJCB 95 (2016) 503-12) and are sites of MT1-MMP accumulation (Monteiro et al. JCB (2013) 1063-79). We have added new data in the

revised manuscript (Fig. 5d) showing that TKS5-positive invadopodia forming in association with nucleus-constricting fibrils in cells invading through the 3D collagen network are associated with collagen cleavage.

In addition, we do not fully agree with the statement that invadopodia form everywhere. Most of the studies have classically used F-actin and cortactin as *bona fide* invadopodia markers. However, these proteins are also the main components of other cellular cytoskeletal structures such as the lamellipodia and it is not always easy to distinguish small lamellipodial and invadopodial structures in a 3D context. With that in mind, we have used the pro-invasive TKS5 protein as an exclusive, lamellipodia-excluded, invadopodia marker (see Fig. 5de). Another potential issue is overexpression of MT1-MMP itself as we noticed that it can lead to MT1-MMP accumulation and some degree of mislocalization at the cell surface. All the experiments related to invadopodia that are reported in Figure 5d-g were performed using MDA-MB-231 cells expressing endogenous MT1-MMP levels.

In addition, the authors do not show whether MT1-MMP only localizes at the nucleolar invadopodia during constricted migration. Gladly, the authors now focus on the product of overall collagenolytic activity, as by performing COL1 3/4 staining in initially migrating cells. Collagen degradation occurs wherever fibrils impinge but –more importantly- constrain the cell body. This takes place at locations of diameter/volume increase, i.e. at pseudopodial bifurcations or in front of the cell nucleus. The authors (anecdotal?) own images in Fig. 1e-h support this simple concept which, however, still needs to be quantified in some way (i.e. measuring the actin/ COL1 3/4 signal along the cell axis of a number of cells, as I already asked in my last review). This way the authors could generate a useful readout. – In contrast, the entire Figure 1 does not show a single structure that I would call an ‘invadopod’. The actin rich spots in Fig. 1a are located at ‘cell-matrix interaction sides’ clarified by the white arrowheads. To name these actin locations ‘invadopodia’ is highly speculative.

R. As suggested by this referee, we have calculated the COL1 3/4 signal along the long axis of cells invading through the collagen gel polymerized at 37 or 20°C. Cells were incubated in the collagen gel for a relatively short period of time (2.5 hrs) to avoid cumulating COL1 3/4 signal, fixed and stained with COL1 3/4 antibodies as described by Wolf and Friedl and line-scan analysis was performed. We generated COL1 3/4 signal profiling by averaging several line-scan profiles obtained from different invasive cells in the 3D collagen gel.

As discussed in the previous point, we did not consider the presence of

invadopodia at this stage of the manuscript and did not incorporate F-actin measurement in the line-scan analysis as F-actin is not specifically associated with invadopodia as a component of other cortical cell structures including lamellipodia.

When cells were incubated in the dense collagen network obtained by polymerization at 37°C, we observed a strong cumulative COL1 3/4 signal associated with the cell rear corresponding to the formation of a collagen tunnel as cells were moving through the collagen gel. Importantly, we found that pericellular collagenolysis was robust in association with the anterior part of the nucleus, while COL1 3/4 signal was minimal at the front edge of the cells (Fig. 1ab). In contrast, collagenolysis was globally reduced when cells were incubated in the permissive collagen gel polymerized at 20°C and there was no such increase of collagenolysis signal ahead of the nucleus under these conditions (Fig. 1ef).

Related to this, as I said before, I believe a connection between Lis and invadopodia, but it does not help the study for the above named reasons. The invadopodia marker stainings in Fig. 4d do not prove at all any MT1-MMP location or collagen degradation during constricted migration.

R. We added new data in this revised version showing that TKS5-positive invadopodia forming in association with constricting collagen fibrils are degradative (Fig. 5d) and that pericellular collagenolysis is increased at the level the bulky region of the cell anterior to the nucleus (Fig. 1a).

In addition, these images have been acquired ON and not IN collagen, which proves the authors statement in the abstract 'whereby mechanical constraints on the nucleus trigger ... invadopodia formation ... in front of the nucleus depending on nucleus-microtubule linkage' wrong. Together, as advice I suggest that the authors exchange their Fig.4d-e with the more correct readout of COL1 3/4 localization of cells IN collagen after the same indicated treatments.

R. The statement regarding the relationship between mechanical constraints on the nucleus and invadopodia formation has been toned down in the abstract and in the text. We would like to emphasize that we show that interfering with Lis1 or Nesprin2/LINC complex function not only affects collagenolysis in the 3D collagen environment (Fig. 5b), but also interferes with invadopodia formation by cells

plated on the fibrous collagen layer (Fig. 5efg). Although in these experiments, cells were plated on top of the fibrous collagen layer, this substratum has some thickness that can be estimated to be ~5-10 μm based on z-stack confocal analysis (some people refer to 2.5D for this type of experimental set-up). We added a simple scheme to illustrate invadopodia formation under these 2D (2.5D) conditions (Fig. 5i of the revised manuscript). We see cells pushing and pulling on collagen fibrils and cells (and nuclei) are squeezed between collagen fibrils as they migrate within this substratum. Thus some level of confinement must exist under these conditions. It is also known from the pioneering work of D. Ingber's lab that mechanical continuity exists between the plasma membrane/cell cortex and the nucleus in cells cultured on a 2D substrate (Maniotis and Ingber PNAS USA (1997) 849-54). Our data point to the existence of a LINC complex, Lis-mediated cortex-nucleus linkage in MDA-MB-231 cells cultured on the 2.5D fibrous collagen layer that is responsible for triggering invadopodia formation at the level of the collagen fibrils, which oppose to cell invasion. We are currently actively investigating relationship between cell pulling and pushing activities, invadopodia formation and collagenolysis using the 2D (2.5D) and 3D systems but we believe that this is beyond the scope of the present study.

2. It is not clear why the authors do not discuss the piston mechanism as it fits wonderfully into their concept. Even though they cite the relevant papers (Refs 64,65) for another reason, they do not reflect on these interesting similar phenomena.

R. We agree that the piston mechanism is conceptually interesting and valid in the context of our study and we are now providing a more detailed discussion as to how this mechanism may be relevant for our model (see Discussion section lines 338-343. Yet, in its original description the piston mechanism implicated actomyosin contractility, while our model is based on microtubule- and Lis1 (probably dynein)-dependent pulling force generation.

3. As said, I generally support the authors concept that the Lis-LINC connection drives nucleus microtubule organization and MT1-MMP localization reflected by reduced invasion rates after Lis knockdown (Fig. 4a). However, this reduction is around 30% compared to a 50% reduction after GM6001 treatment (Fig. 1a). The authors should therefore discuss possible compensatory effects, i.e. by other nucleo-cytoskeletal connections i.e. actin, or others.

R. This point is well taken and possible compensatory mechanisms are now

discussed in the Discussion Section (lanes 334-346).

Also, as an important final control, is the Lis defect restored in collagen with wide pores?

R. Fig. 5a shows that Lis1 silencing does not affect the invasion speed of MDA-MB-231 cells in the large pore collagen environment.

4. The authors change lamin A levels by downmodulation or overexpression. It has been previously shown that lamin A changes modulate nuclear stiffness. The authors, however, do not show it in this manuscript. They should therefore be careful with their statements (as mentioned in my last review). Instead of stating in the abstract 'Here, we show that modulation of matrix pore size or nuclear stiffness directly impinges on levels of MT1-MMP-mediated pericellular collagenolysis by cancer cells' the statement needs to be like 'Here, we show that modulation of matrix pore size or lamin A expression known to modulate nuclear stiffness directly impinges on levels of MT1-MMP-mediated pericellular collagenolysis by cancer cells'. This has to be revised through the entire text.

R. We agree. Changes have been introduced accordingly throughout the text.

Other points

1. GM6001 is a general MMP inhibitor, and the authors should refer to GM6001 as such (in lanes 104, 106).

R. The text has been changed to refer to GM6001 as a general MMP inhibitor. We also introduced data showing the effect of MT1-MMP silencing on collagenolysis side-by-side to GM6001 data in Fig. 1d of the revised manuscript.

2. The presentation of results sometimes appears fractionated and results could be summarized:

(A) i.e. Figures 1k and S1b could be merged to one figure and shown next to Fig. 1j - the reader would understand better the interesting point that migration in 20C collagen is fast despite only little proteolysis which is an important point guiding the reader to the central dogma of this study. The measurement of the proteolysis level in the presence of GM6001 is missing.

R. Figure 1k and Figure S1B are now displayed side-by-side in Figure 1g of the revised manuscript close to data on invasion speed. Data showing the effect of GM treatment and MT1-MMP silencing on collagenolysis have been included (Figure 1d).

(B) Figs 2A and S2c,d belong together and it is a pity that results were acquired in different migration systems. Perhaps the authors could think of a way to summarize the results into one figure.

R. The main purpose of the data shown in Fig. S2cd was to show that modulation of lamin A levels by downmodulation or overexpression in MDA-MB-231 cells could affect nucleus speed through microfluidic constriction channels probably as a result of modulation of nucleus deformability and nuclear stiffness as discussed in Rowat *et al.* J. Biol. Chem. (2013) 288:8610-8618.

In this regard, in Fig. S2d, the authors should find a way to quantify the migration speed of GFP-LMNA cells even if it is very slow/ nucleus eventually stalls.

R. Migration speed of GFP-LMNA cells/nuclei has been determined and is described in Fig. S2D in the revised manuscript ($\leq 0.02 \mu\text{m}/\text{min}$, n=2 cells, 11 cells stalled)

(C) Figs. 3i and S4a belong together and should be supported by corresponding image examples.

R. We understand the point raised by the referee. However, based on earlier comments from the reviewers regarding a previous version of the manuscript we had made efforts avoiding breaking figure order in the text. In this respect, data reporting the effects of inhibition of LINC complex function using DN-KASH expression have all been grouped (see Figure 3 and supplemental Figure 4), while data regarding Lis1/Nesprin-2 are displayed in Figure 4 and 5 (and supplemental Figure 5) of the revised manuscript. As requested by the referee, images showing DAPI and pericentrin immuno-staining are displayed in supplemental Figure 4d (DN-KASH/KASHext) and in Figure 4f (siNesprin-2 and siLis1).

3. The time annotation in Figs. 2e,h,k and 3a does not fit to their corresponding movies.

R. We apologize for discrepancies between time annotations in the figures and corresponding movies. Time annotations in revised Fig. 2h and k and Fig. 3c have been corrected. Time annotations in Fig. 2e were correct.

4. In Fig. S1a, the nuclear morphology showing a 'prolapse' rather looks like a nucleus with a micronucleus. I recommend to take it out and exclude such images from analysis.

R. We agree that this image was not optimal and it has been replaced by a better one in the revised manuscript (Fig. S1a). However, we would like to emphasize that nucleus morphology data have been obtained by analyzing 3D confocal stacks of images (images shown in Figure S1a are single optical sections taken from these stacks) allowing visualizing whether prolapse was connected to the nucleus, while micronuclei clearly separated from the nucleus were systematically discarded from the analysis.

5. There are text passages that do not read entirely accurately, i.e.:

R. We thank the referee for his/her suggestions.

-Lane 83 'F-actin enrichments were visible ahead of the nucleus in association with ECM fibrils oriented orthogonally to the direction of migration, probably opposing cell movement' Reads as if F-actin opposes cell movement

R. This sentence has been deleted in the revised version.

-lane 110 'All together, these findings indicated that surface-exposed MT1- MMP enabled confined nuclear and cell movement' reads as if MT1-MMP drives nuclear movement which the authors did not really investigate and which i.e. also appears in the absence of MT1-MMP activity (Fig. 1c). A better sentence here could be: 'All together, these findings indicated that surface-exposed MT1- MMP enabled efficient cell movement in confinement by mediating proteolysis anterior of the nucleus.'

R. The sentence has been changed according to the referee's suggestion (lane 98-100).

-lane 119 'In the large pore size collagen network, the cell body was generally larger and more massive and the nucleus was positioned close to the cell center (Fig. 1i)'. For such a statement the authors need add some quantification.

R. This sentence has been deleted.

-lane 311 The statement in the discussion 'Our data reveal an unprecedented mechanotransduction pathway in which tumor cells adapt to the 3D matrix environment by sensing constraints imposed on the nucleus and...'. Can the pheomenon described here really be called a mechanotransduction pathway which typically involves signaling molecule activation by some means of mechanostimulation? The authors may tone down this statement to i.e. 'Our data reveal a novel mechanical mechanism in which tumor cells adapt to the 3D matrix environment by sensing constraints imposed on the nucleus and...'

R. We agree. The sentence has been changed (lanes 308-310).

-lane 322, reads confusing 'The present study suggests that nucleus pulling force generation and the multistep invadopodia assembly program may be two sides of the same coin, governed by nucleus-cortex mechanical coupling through the microtubule network'. Pulling forces should be generated by the actin cytoskeleton, or better specified what pulling forces mean in the context of the microtubulus network.

R. This sentence has been deleted.

Reviewer #2 (Remarks to the Author):

The revision of this paper from Infante et al. is improved. As I said before, I feel that the paper is largely acceptable for publication without the addition of any further experimentation. However, the way that paper is presented makes it quite tough reading. In particular, I think that the novelty of the main story is somewhat obscured by the inclusion of data that confirm previous findings and this is maybe why one of the other reviewers is struggling with the paper. Furthermore, some of the figure formatting and presentation could be improved to render the paper easier to read and thus increase its impact.

R. We thank the reviewer for his/her positive comments and for his/her support to improve the manuscript. We acknowledge the fact that some data displayed in Figure 1 are confirmatory of some published information. We do believe, however, that these data, obtained using our own experimental set-up, are worth to be displayed as a starting point for the present study. Yet, to take into account this referee and referee's #1 comments, Figure 1 in the revised manuscript has been modified to include new data documenting and quantifying collagenolysis more accurately.

Specific comments on presentation:

1. Fig. 1g: I feel that the inserts are inadequate as they stand as their resolution is insufficient. If the point of these is to show association of col13/4 with actin, it would be preferable to show an actual quantification – or a much higher resolution image – or both.

R. These data have been deleted in the revised version of the manuscript. Figure 1 now includes new data showing the quantification of pericellular COL1 3/4 (collagenolysis) signal along the cell axis through line-scan analysis. We generated averaged COL1 3/4 signal profiling based on several invasive cells in the 3D collagen that revealed that pericellular collagenolysis was increased in association with the anterior part of the nucleus, while COL1 3/4 signal was minimal at the front edge of the cells incubated in the confining 3D collagen environment (Fig. 1ab). In contrast, collagenolysis was reduced when cells were incubated in the permissive collagen gel polymerized at 20°C and there was no such increase ahead of the nucleus under these conditions (Fig. 1ef).

2. I feel that all of figure 1 would be better relegated to supplementary. This is because, most – if not all – of these observations have been published before in one way or another, and this is if perhaps why one of the other reviewers is struggling with seeing the novelty of this paper. A better first figure would be to present the different conditions showing the links between confinement, nuclear deformation and invasion.

This would make a better exposition of the key rationale for the rest of the paper.

R. As discussed above, we believe that the description of these data in Fig. 1 makes a strong starting point for the study and novelty in this figure stands in the fact that degradation is reduced in the presence of collagen polymerized at 20°C while invasion is maintained. Figure 1 has been reorganized to emphasize this point.

3. Fig 3g would be more convincing if there were some colocalisation demonstrated between Lis 1 and a nuclear envelope marker.

R. We are providing new data in Fig. 4c of the revised manuscript showing colocalization of Lis1 with the outer NE protein Nesprin-2.

4. GM6001 is not an specific MT1-MMP inhibitor – especially at 40µM. This assertion needs to be corrected in the text and mention made of the experiments performed with siRNA of MT1-MMP.

R. The text has been changed to refer to GM6001 as a general MMP inhibitor. We also introduced data showing the effect of MT1-MMP silencing on collagenolysis side-by-side to GM6001 data in Fig. 1d of the revised manuscript.

5. Should the x-axis of Fig. S1f have a label – say, ‘interfibrillar distance’

R. X-axis has been properly labeled.

6. I don’t understand why the data showing nesprin2 displacement after DN-KASH expression is a a format that differs from that used to display the nesprin1 data (sup. Fig. 4b &c).

R. Data showing Nesprin-2 displacement in DN-KASH expressing cells (revised Fig. S4c) is now provided in the same format as for Nesprin-1 (Fig. S4a) and SUN1 (Fig. S4c).

Also, what is Fig. S4d attempting to show? – it appears quite confusing to me.

R. This panel has been deleted.

7. Where is the GFP in Fig. S6?

R. Data initially shown in Fig. S6 is now displayed in Fig. S4gh of the revised manuscript in which we grouped all supplemental data related to DN-KASH expression. As a control construct for these experiments, we used GFP fused with the KASHext peptide that does not interact with SUN proteins and does not interfere with Nesprin-1, -2 or SUN-1 association with the NE (Fig. S4abc).

Reviewer #3 (Remarks to the Author):

Please note that reviewer 3 only commented for the editors and still finds the advance of this manuscript limited.

REVIEWERS' COMMENTS:

Reviewer #1 (Remarks to the Author):

The manuscript Infante et al. on the mechanism of nucleus-mediated anterior MT1-MMP endosome localization has improved to some extent since its last submission and is, all together, on a good way. However, there are a number of remaining points of criticism/ options for improvement:

1. There are still some text passages that read hard, or appear somewhat preliminary, including newly inserted text passages and figures. A number of examples are given below.
2. To show sufficient advancement from the previous literature at the beginning of the manuscript, as reviewer #2 points out as well, I suggest to re-arrange Figure 1 (specified below).
3. I am still not really convinced by the invadopodia concept of the authors at the end of the manuscript, as I do not understand how invadopodia should form upon confinement/ pressure, and the authors do not really present an explanation. I believe the actual data in the manuscript, but ending with a '2.5D' approach on collagen (Fig. 5e,f) rather weakens the manuscript, when 3D confinement is critical for the observed effects throughout the entire manuscript. I had mentioned this in some form already in my last review. To make their point and round up the study, the authors should, after showing the control condition in Fig. 5d, end by testing collagenolysis of cells (along TKS5 positive structures; these new data in Fig. 5d look quite convincing) in 3D confining collagen after transient Lis / nesprin inhibition. Collagenolysis should serve as readout for disturbed nucleus-mediated endosome steering the same way as shown in Fig. 1b,f. I recommend to move Fig. 5e,f to the supplement as these data are based on 2.5 D assays. Even if some cells may 'invade' into this 5-10 um thin layer, this phenomenon is probably quite anecdotal and the 2.5D matrix in this regard a rather poorly controllable model. All together, I recommend that the authors discuss the invadopodia concept as long as it remains toned down and rather underpin this potentially interesting phenomenon in their ongoing 2D/3D study.

Some of the figures do not really appear smooth yet:

-Figure 1 has the potential to nicely delineate the advance from previous studies, however, is hard to read and still appears somewhat preliminary. With some re-adjustments the figure could profit a lot and the novelty could be fleshed out better. In agreement with reviewer #2, I strongly recommend to place directly next to each other or underneath each other invasion speed, collagen degradation and nuclear deformation of MDA-MD-231 cells, both in the absence and presence of GM6001 and for the 37C and 20C induced collagen pore size. As mentioned in my last review, it is important to add the missing data set on collagen degradation in 20C polymerized collagen in the absence and presence of GM6001. This way the reader can immediately appreciate the context of invasion, collagen degradation and nuclear deformation. Accompanying panels, such as h and i could be moved into the supplement. In addition, move panels from Figure 2a-d to the end of Figure 1. Figure 1 could then end with the new main message that confinement, induced by both pore size and lamin A mediated nuclear rigidity, determines whether a cell requires nucleo-anterior proteolysis for migration. The beautiful visualization of MT1-MMP endosome steering by the nuclear 'piston', which only works in confinement, will then be shown by the remaining part of Figure 2.

-Figure 1b: Numbering in 'Distance to nuclear center' appears inaccurate. The figure would benefit from merging with Fig. 1f, to recognize the impact of pore size on distribution of collagen degradation signal. The rather smoothly decreasing (where cell rear is at the left and leading edge on the right side I suppose) orange curve does not really agree with the image in Fig. 1a (see white arrows) where the degradation signal is clearly enhanced in front of the nucleus. In addition, the pastel-colored area, marking 'Degradation anterior to the nucleus' may not be located entirely accurately as it ends (too far on the left side) where the nucleus center should be (the x-axis is not entirely clear).

-Figure 3a,b: it appears somewhat unlogical why the authors use here GM6001 instead of a

condition of large pore size (polymerization at 20C), as has been used as the working model in Figs. 2 and 5.

Figure 5c: numbers of cells and endosomes in one of the conditions are missing.

Examples for text inaccuracies

-line 57; MT1-MMP surface exposure, better substitute with ...surface action or localization

-line 84; 'a collagen tunnel' – this does not read correct to me when collagenases clear collagen from the cell path.

-in Fig. S3b, the altered nuclear morphologies after lamin A KD of a cell plated on collagen are quite striking. These should be at least mentioned/ discussed, as the morphological alterations possibly contribute to an incomplete 'piston' mechanism which keeps MT1-MMP endosomes anterior to the nucleus.

-line 158 '... and surface exposure of active MT1-MMP...' where have you shown this?

-line 254 '...positioned behind the nucleus...' is not entirely accurate; it is better to state that the centrosome is only at 10% of the time positioned in front of the nucleus.

-line 314. Could you extend a bit more on how the microtubule-centrosome system exerts pulling forces onto the nucleus?

-line 344: '... interesting similar mechanism...' . In my opinion it is the same mechanism (in the piston mechanism, the nucleus is pushed forward under conditions of confinement and pushes MT1-MMP endosomes as well as creates some overpressure, and this all by combined action of actin and tubulin structures). The authors should acknowledge that they are looking at a different molecular system assisting the same phenomenon, which is on the consequences of nuclear forward movement during migration in confinement.

-line 787: left and right axis are mixed up

Reviewer #2 (Remarks to the Author):

The paper is acceptable for publication.

I have the following minor comments/suggestions:

1. In the new Figs. 1b and 1f, the difference in mean collagen degradation between 1b and 1f is not convincing. The graphs in Figs. 1c, d, g, h, and f are much more convincing. Can the authors not remove 1b and 1f? i don't think that they help much.

2. Fig. 1 is badly organised and so difficult to follow. Is it possible for the authors to think about how they assign the figure call-outs so that the figure is more logically organised.

3. In 5c, there are number missing on the right hand panels - see c:? e:? should there not be numbers after the colons?

Re : NCOMMS-17-06498B

Reviewer #1 (Remarks to the Author):

The manuscript Infante et al. on the mechanism of nucleus-mediated anterior MT1-MMP endosome localization has improved to some extent since its last submission and is, all together, on a good way. However, there are a number of remaining points of criticism/ options for improvement:

1. There are still some text passages that read hard, or appear somewhat preliminary, including newly inserted text passages and figures. A number of examples are given below.

Examples for text inaccuracies

-line 57; MT1-MMP surface exposure, better substitute with ...surface action or localization

R./ Line 57 changed to : MT1-MMP surface localization

-line 84; 'a collagen tunnel' – this does not read correct to me when collagenases clear collagen from the cell path.

R./ Lines 84-85 changed to : ... and cleared collagen from the cell path probably through the action of collagenases, consistent with previous observations

-in Fig. S3b, the altered nuclear morphologies after lamin A KD of a cell plated on collagen are quite striking. These should be at least mentioned/ discussed, as the morphological alterations possibly contribute to an incomplete 'piston' mechanism which keeps MT1-MMP endosomes anterior to the nucleus.

R./ Line 145-146 : the altered nuclear morphology in LMNA-depleted cells plated on a thick fibrous collagen layer is now mentioned.

The piston mechanism in relation with nuclear stiffness, nuclear forward movement and endosome polarization is discussed in the Discussion section (lines 451-469).

-line 158 ‘... and surface exposure of active MT1-MMP...’ where have you shown this?

R./ Lines 158-160 changed to : All together, these findings indicated that MT1-MMP-mediated pericellular collagenolysis is an adaptive response that is switched on during confined migration in the dense ECM environment, suggesting a relationship between environment and cell (nuclear) biomechanics and modulation of active MT1-MMP at the cell surface.

-line 254 ‘...positioned behind the nucleus...’ is not entirely accurate; it is better to state that the centrosome is only at 10% of the time positioned in front of the nucleus.

R./ Lines 260-262 changed to : ..., we found that the centrosome was at ~50% of the time positioned ahead of the nucleus in control cells in agreement with data described above (Fig. 4g). Conversely, the centrosome was only at 15% of the time positioned in front of the nucleus in cells knocked down for Lis1 (Fig. 4g).

-line 314. Could you extent a bit more on how the microtubulue-centrosome system exerts pulling forces onto the nucleus?

R./ Line 242-244: We added a sentence to describe the prevailing model of nucleo-centrosome attachment.

.../... Dynein motor and its regulator Lis1, which is essential for high-load dynein functions, have been implicated in nucleus-centrosome linkage during neuronal migration³⁴⁻³⁷. The prevailing model is that SUN-nesprin1/2 complexes mediate nucleo-centrosome coupling providing anchors to cytoplasmic dynein/Lis1 complexes to pull the nucleus toward the centrosome³⁸ .../...

-line 344: ‘... interesting similar mechanism...’ . In my opinion it is the same mechanism (in the piston mechanism, the nucleus is pushed forward under conditions of confinement and pushes MT1-MMP endosomes as well as creates some overpressure, and this all by combined action of actin and tubulin structures). The authors should acknowledge that they are looking at a different molecular system assisting the same phenomenon, which is on the consequences

of nuclear forward movement during migration in confinement.

R./ The discussion section has been reworded to explain better potential relationship of the mechanism we describe here and the piston mechanism (lines 353-360).

-line 787: left and right axis are mixed up

R./ The legend of Fig. 1b has been corrected.

2. To show sufficient advancement from the previous literature at the beginning of the manuscript, as reviewer #2 points out as well, I suggest to re-arrange Figure 1 (specified below).

R./ Suggested changes to Figure 1 have been introduced.

Some of the figures do not really appear smooth yet:

-Figure 1 has the potential to nicely delineate the advance from previous studies, however, is hard to read and still appears somewhat preliminary. With some re-adjustments the figure could profit a lot and the novelty could be fleshed out better.

In agreement with reviewer #2, I strongly recommend to place directly next to each other or underneath each other :

invasion speed

collagen degradation

nuclear deformation of MDA-MD-231 cells

both in the absence and presence of GM6001 and for the 37C and 20C induced collagen pore size.

R./ We thank the referee for his/her suggestion to improve the clarity of Figure 1 and Figure 2. Nuclear deformation , invasion speed and collagen degradation in 37°C vs. 20°C collagen \pm GM data are now shown in Fig. 1d to f.

As mentioned in my last review, it is important to add the missing data set on collagen degradation in 20C polymerized collagen in the absence and presence of GM6001.

This way the reader can immediately appreciate the context of invasion, collagen degradation and nuclear deformation.

R./ Done. 37°C vs. 20°C ± GM data are now shown in Fig. 1f.

Accompanying panels, such as h and i could be moved into the supplement. In addition

R./ Done. HT-1080 data are now shown the Supplementary Figure 1g and h.

move panels from Figure 2a-d to the end of Figure 1. Figure 1 could then end with the new main message that confinement, induced by both pore size and lamin A mediated nuclear rigidity, determines whether a cell requires nucleo-anterior proteolysis for migration.

R./ Done. LMNA data are now shown in Figure 1h to l.

-Figure 1b: Numbering in 'Distance to nuclear center' appears inaccurate.

The figure would benefit from merging with Fig. 1f, to recognize the impact of pore size on distribution of collagen degradation signal.

The rather smoothly decreasing (where cell rear is at the left and leading edge on the right side I suppose) orange curve does not really agree with the image in Fig. 1a (see white arrows) where the degradation signal is clearly enhanced in front of the nucleus.

In addition, the pastel-colored area, marking 'Degradation anterior to the nucleus' may not be located entirely accurately as it ends (too far on the left side) where the nucleus center should be (the x-axis is not entirely clear).

R./ Figure 1b and 1f have been merged to give Fig. 1b.

Enhancement of collagen degradation ahead of the nucleus in 37C vs 20C polymerized collagen is now highlighted by merging intensity profiles in 37C vs 20C polymerized collagen in revised Fig. 1b. Rear and Front positions are indicated.

To average intensity profiles from several cells (n=24) with variable position of the nucleus, the software symmetrizes the "rear" and "front" halves of each cell assigning to the nucleus center the "0" position in the x-axis. Since the nucleus is generally located closer to the cell rear, symmetrizing generates some distortion

due to stretching of the rear "half" of the cells.

The beautiful visualization of MT1-MMP endosome steering by the nuclear 'piston', which only works in confinement, will then be shown by the remaining part of Figure 2.

R./ Data regarding MT1-MMP endosome polarization is now shown in Figure 2.

3. I am still not really convinced by the invadopodia concept of the authors at the end of the manuscript, as I do not understand how invadopodia should form upon confinement/ pressure, and the authors do not really present an explanation. I believe the actual data in the manuscript, but ending with a '2.5D' approach on collagen (Fig. 5e,f) rather weakens the manuscript, when 3D confinement is critical for the observed effects throughout the entire manuscript. I had mentioned this in some form already in my last review. To make their point and round up the study, the authors should, after showing the control condition in Fig. 5d, end by testing collagenolysis of cells (along TKS5 positive structures; these new data in Fig. 5d look quite convincing) in 3D confining collagen after transient Lis / nesprin inhibition. Collagenolysis should serve as readout for disturbed nucleus-mediated endosome steering the same way as shown in Fig. 1b,f. I recommend to move Fig. 5e,f to the supplement as these data are based on 2.5 D assays. Even if some cells may 'invade' into this 5-10 um thin layer, this phenomenon is probably quite anecdotal and the 2.5D matrix in this regard a rather poorly controllable model. All together, I recommend that the authors discuss the invadopodia concept as long as it remains toned down and rather underpin this potentially interesting phenomenon in their ongoing 2D/3D study.

R./ We respectfully disagree and we would like to maintain data showing the relationship between TKS5-positive structures and collagenolysis and Lis1 and nesprin-2 proteins. In addition, new data have been added (see new Fig. 5e) indicating that concomitant to the global reduction of pericellular collagenolysis (Fig. 5b), nucleo-anterior collagenolysis was reduced in cells knocked down for Lis1 or Nesprin-2 in 3D type I collagen gel polymerized at 37°C (Fig. 5e). The model of the adaptive collagenolysis response has been moved to a new figure

6.

As requested by this referee we toned the discussion regarding invadopodia in the Result Section (lines 355-369), in which we replaced the term “invadopodia” by “TKS5-positive structures”, the structures that were scored to generate data in Fig. 5d-h.

-Figure 3a,b: it appears somewhat unlogical why the authors use here GM6001 instead of a condition of large pore size (polymerization at 20C), as has been used as the working model in Figs. 2 and 5.

R./ The rationale of this experiment was to analyze cells in situation of confinement in the 37C polymerized gel in the presence of MT1-MMP inhibitor.

Figure 5c: numbers of cells and endosomes in one of the conditions are missing.

R./ Cell and endosome number have been added

Reviewer #2 (Remarks to the Author):

The paper is acceptable for publication.

I have the following minor comments/suggestions:

1. In the new Figs. 1b and 1f, the difference in mean collagen degradation between 1b and 1f is not convincing. The graphs in Figs. 1c, d, g, h, and f are much more convincing. Can the authors not remove 1b and 1f? i don't think that they help much.

R./ To better highlight existing differences between collagen degradation in 37°C vs 20°C collagen, 3/4C intensity profiles in gels polymerized at 37°C or 20°C have been merged in revised Fig. 1b. Accumulation of cleaved collagen ahead of the nucleus can be observed in several images shown in Fig. 1a (panel iii), in Fig. 5d and Fig. 5e (upper panel, siNT).

2. Fig. 1 is badly organised and so difficult to follow. Is it possible for the authors to think about how they assign the figure call-outs so that the figure is more logically organised.

R./ On request of this referee and referee #1, Figure 1 has been reorganized.

3. In 5c, there are number missing on the right hand panels - see c:? e:? should there not be numbers after the colons?

R./ We apologize, missing cell and endosome numbers in Fig. 5c have been added.